# `FHAIM`: Fully Homomorphic AIM for Private Tabular Synthetic Data Generation

Mayank Kumar [1]   Qian Lou [1]   Paulo L. Barreto [2]   Martine De Cock [2]   Sikha Pentyala [2]

## Abstract

Data is the lifeblood of AI, yet much of the most valuable data remains locked in silos due to privacy and regulations. As a result, AI remains heavily underutilized in many of the most important domains, including healthcare, education, and finance. Synthetic data generation (SDG), i.e. the generation of artificial data with a synthesizer trained on real data, offers an appealing solution to make data available while mitigating privacy concerns, however existing SDG-as-a-service workflow require data holders to trust providers with access to private data. We propose `FHAIM`, the first fully homomorphic encryption (FHE) framework for training a marginal-based synthetic data generator on encrypted tabular data. `FHAIM` adapts the widely used AIM algorithm to the FHE setting using novel FHE protocols, ensuring that the private data remains encrypted throughout and is released only with differential privacy guarantees. Our empirical analysis show that `FHAIM` preserves the performance of AIM while maintaining feasible runtimes. Code: [https://github.com/mayank64ce/mbi](https://github.com/mayank64ce/mbi)

## 1. Introduction

Despite the fact that modern AI applications are heavily data-driven, much of the most valuable real-world data remains siloed within research centers, hospitals, companies, and financial institutions. These silos are typically protected by strict access controls and regulations due to the sensitive and personal nature of the data they contain. This, while necessary, significantly hampers data sharing and reuse. As a result, AI remains underutilized in many high-impact domains, including healthcare, genomics, biomedicine, commerce, education, and finance.

*Synthetic data generation (SDG)*, i.e., the process of generating artificial data using a synthesizer trained on real data, offers an appealing approach to facilitate data sharing while mitigating privacy concerns (Hu et al., 2024). When done well, synthetic data has the similar characteristicsas the original data but, crucially, without replicating personal information. This has led to SDG becoming a widely adopted tool for enabling data access (NSF News, 2023; FDA, 2023). This success has facilitated numerous SDG-as-a-service platforms[1] where data holders outsource the process of SDG to third parties.

While many SDG-as-a-service platforms are marketed as privacy-preserving, they typically provide at most only *output privacy*, most commonly through techniques such as differential privacy (DP) (Dwork et al., 2006) which aim to ensure that the released synthetic data does not leak information about individual records in the real data that was used to train the generator.[2] However, these approaches implicitly assume that the service provider (the third party) has full access to the raw training data in plaintext. In many practical settings, data holders are subject to strict privacy regulations and organizational constraints that prohibit disclosure of sensitive data to third parties, as the risks associated with personal data exposure can be harmful (Martin, 2017; Thompson & Warzel, Dec 19, 2019). *Input privacy*, i.e. protecting the confidentiality of the input data, remains an open problem in outsourced synthetic data generation.

While approaches based on secure multiparty computation (Pentyala et al., 2024) and federated learning (Maddock et al., 2024) offer forms of input privacy, they require coordination among multiple non-colluding computational parties, making them unsuitable for SDG-as-a-service deployed *by a single service provider for a single data holder*. To our knowledge, no prior work addresses providing input privacy in such settings.

We propose to leverage fully homomorphic encryption

---

[1]University of Central Florida [2]University of Washington Tacoma. Correspondence to: Sikha Pentyala <sikha@uw.edu>, Mayank Kumar <mayank.kumar@ucf.edu>.

*Proceedings of the 43rd International Conference on Machine Learning*, Seoul, South Korea. PMLR 306, 2026. Copyright 2026 by the author(s).

[1]Examples include Mostly AI, Gretel (acquired by NVIDIA), Syntegra, MDClone, Hazy, and Tonic.

[2]DP substantially improves the robustness of synthetic data against adversarial attacks (Golob et al., 2025).

| Approach | Input Privacy | Output Privacy | #Data Holders | #Computing Parties |
|---|---|---|---|---|
| AIM (McKenna et al., 2022) | × | Global DP on unencrypted data | 1 | 1 |
| CAPS (Pentyala et al., 2024) | MPC | Global DP in MPC protocols | 1 or Multiple | Multiple* |
| FLAIM (Maddock et al., 2024) | FL | Distributed DP on unencrypted data | Multiple | Multiple |
| FHAIM (ours) | FHE | Global DP in FHE protocols | 1 | 1 |

*Table 1.* **Input Privacy Extensions for AIM.** *: needs more than 1 non-colluding computing server, efficient with 3.

(FHE) (Gentry, 2009; Rivest et al., 1978) to enable synthetic data generation in outsourced settings, presenting the first FHE-enabled framework for SDG. Our approach allows an SDG-as-a-service provider to train synthetic data generators directly on encrypted data without ever accessing the raw data (data holders encrypt their data before providing it to the service provider), thus providing input privacy. We focus on tabular data, a widely used data modality in privacy-sensitive domains such as healthcare, finance, and public administration.

To demonstrate the feasibility of our approach, we propose FHAIM, built on AIM (McKenna et al., 2022), a state-of-the-art method for DP tabular data generation (Chen et al., 2025; Tao et al., 2022). AIM is a marginal-based method that fits a joint probability distribution on the real data and subsequently samples from it to generate synthetic data. AIM does this in an iterative manner: in each iteration (1) a subset of attributes is selected for which the marginal probability distribution over the real data differs the most from the current version of the synthetic data (*select step*); (2) this marginal distribution is estimated over the real data (*measure step*); and (3) this estimated marginal is added to a probabilistic graphical model and a new version of the synthetic data is generated (the *generate step*). To provide formal output privacy with DP, the select step is non-deterministic, with attribute subsets that yield a higher mismatch more likely to be chosen; likewise, in the measure step, Gaussian noise is added to the marginal probability measurements.

The key idea of FHAIM is to perform all data-dependent operations in the select and measure steps within FHE. We design novel FHE protocols for these operations, effectively implementing DP within FHE (DP-in-FHE), meaning that the outputs of the FHE operations are protected with DP noise, even after decryption. Specifically, the data holder provides the service provider with an encrypted version of their private dataset, along with encrypted unit noise samples. This enables the service provider to perform the data-dependent operations in the select and measure steps on the encrypted data. Once the noisy marginals are obtained, they are decrypted, and the provider proceeds with the generation step in-the-clear, since this step is data-independent and operates only on the already-privatized statistics. Importantly, the service provider never observes the noise used to provide DP guarantees. Hence, in addition to never seeing the real data, the service provider never observes the probability distributions estimated on the real data, thus providing stronger privacy guarantees. Our main contributions are:

- **First FHE-based SDG framework.** We introduce FHAIM, the first system that enables training marginal-based synthetic data generators directly on fully homomorphically encrypted data, providing input privacy without requiring multiple non-colluding parties.
- **Novel DP-in-FHE protocols.** We design efficient FHE protocols for marginal computation ($\pi_{\text{COMP}}$), differentially private query selection ($\pi_{\text{SELECT}}$), and noisy measurement ($\pi_{\text{MEASURE}}$), implementing the Gaussian and exponential mechanisms entirely within the encrypted domain. Towards this, we propose an efficient encrypted memory layout that enables marginal computation with multiplicative depth depending only on marginal degree $k$, ensuring scalability. We also propose to replace the $L_1$-norm quality score with a squared $L_2$-norm in the select step, avoiding unstable polynomial approximations of the absolute value function in FHE.

We demonstrate practical feasibility on three real-world datasets, achieving runtimes of $\sim$11 to $\sim$30 minutes while preserving the statistical utility and downstream ML performance of the original AIM algorithm.

## 2. Related Work

There is ample work on SDG with output privacy; see (Du & Li, 2025) and references therein. The approach that we develop in this paper for AIM, can be extended to other SDG algorithms such as MST (McKenna et al., 2021) and RAP (Vietri et al., 2022).

In contrast to the wealth of literature on SDG with formal output privacy guarantees, existing research on SDG with input privacy is very limited. (Pentyala et al., 2024) leverage secure multiparty computation (MPC) to provide input privacy. While designed for scenarios with multiple data holders, e.g., multiple hospitals who want to train a generator over their combined data, an MPC setup can in principle be used for the single data holder scenario that we consider in this paper. Indeed, the data holder could send encrypted shares of their data to a set of MPC servers, who then proceed to train the generator. But MPC has a

strong non-collusion assumption: if the computing servers collude, they can reconstruct the original data. As such, it may be difficult to set up a credible SDG-as-a-service based on MPC in the cloud of a single provider. Indeed, if all computing servers in an MPC setting reside with the same provider or entity (e.g. a federal agency), then the risk and the perception that they may collude and reconstruct the original data may be high.

Federated Learning (FL) based approaches, such as FLAIM (Maddock et al., 2024), provide input privacy in distributed, multi-silo settings by ensuring that raw data never leaves each data holder. This is achieved by letting each client perform the select and measure computations on site, and only share (privatized) statistics with a central server who updates a global probabilistic model that can be accessed by all the data holders. This approach, which is designed for multiple data holder scenarios, does not lend itself to the single service provider, single data holder scenario that we consider in this paper. Even in a hypothetical FL setup with only one data holder, that data holder would be performing a lot of the computations, going against the intent of outsourcing the SDG training to a service provider.

Table 1 summarizes the distinction between our proposed approach and the closest existing work. To the best of our knowledge, there is no work on training of synthetic data generators over data that is encrypted with fully homomorphic encryption (FHE), neither with AIM nor with any other SDG algorithm for any kind of data modality. Existing research on combining FHE and DP to compute DP statistics such as marginals (Roy Chowdhury et al., 2020; Ushiyama et al., 2022; 2021; Bakas et al., 2022) focuses on individual statistics rather than SDG training. For completeness, we mention that while recent work explored "DP for free" from HE noise (Ogilvie, 2024), this faces significant barriers (data-dependent variance, noise growth, parameter-dependence) and remains largely theoretical. Our approach explicitly considers DP noise in FHE for standard DP guarantees and can be adapted to future progress in this area.

## 3. Preliminaries

### 3.1. Fully Homomorphic Encryption

Given a target plaintext transformation $f(\cdot)$ that maps an input $x$ to an output $y$, fully homomorphic encryption (FHE) defines a corresponding homomorphic function $g(\cdot)$ that operates on the encrypted input $\text{Enc}_{pk}(x)$. The result of this operation remains in the ciphertext domain, satisfying the core consistency equation: $\text{Dec}_{sk}(g(\text{Enc}_{pk}(x))) = f(x)$ where $\langle pk, sk \rangle$ are a mathematically linked *public key* and *secret key* pair. The cryptographic security of FHE schemes relies on the Learning With Errors (LWE) assumption (Regev, 2009). Practical implementations frequently utilize the Ring

LWE (RLWE) (Lyubashevsky et al., 2013) variant, which is structured over the polynomial ring $R_Q = \mathbb{Z}_Q[x]/\langle x^N + 1 \rangle$ with prime modulus $Q$. In this setting, the polynomial degree $N$ is a critical security parameter, typically selected within the range of $2^{12}$ to $2^{16}$. Note that in the following sections, $[\![x]\!] = \text{Enc}_{pk}(x)$.

FHE facilitates computation on encrypted data through a suite of fundamental primitives. Specifically, homomorphic addition (HE.Add) and homomorphic multiplication (HE.Mult) execute arithmetic operations on pairs of ciphertexts, while homomorphic rotation (HE.Rot) performs cyclic shifts on encrypted vectors. Correctness of FHE operations is guaranteed provided two conditions are met: (1) the ciphertext level remains within the predetermined multiplication depth limit, and (2) the accumulated noise does not exceed the decryption radius (typically $Q/2$ for the ciphertext modulus $Q$). Under these constraints, decryption maintains accuracy within a prescribed error tolerance, typically $10^{-9}$ to $10^{-12}$.

In this paper, we employ the CKKS scheme (Cheon et al., 2017), which supports approximate arithmetic on real numbers. While the computation of marginals (i.e., counts) in AIM is integer-valued, subsequent operations such as the noise addition to provide DP guarantees involve real-valued functions. Though integer based schemes such as BFV (Fan & Vercauteren, 2012) and BGV (Brakerski et al., 2014) are efficient for marginals computations, they would require costly fixed-point encoding and rescaling operations for each real-valued computation. This makes CKKS the most efficient choice for our work with DP-in-FHE. In Section 6 we compare the utility of synthetic data generated with FHAIM (over encrypted data) with AIM (over plaintext data), empirically demonstrating that the use of an approximate FHE scheme has minimal impact on the results.

### 3.2. Differential Privacy

Differential Privacy (DP) is a formal privacy notion that bounds the impact an individual in a dataset $D$ can have on the output of an algorithm or operation $\mathcal{A}$ performed over $D$ (Dwork et al., 2006). Formally, $\mathcal{A}$ is called $(\varepsilon, \delta)$-DP if for all pairs of neighboring datasets $D, D' \in \mathbb{D}$, i.e., $D'$ can be obtained from $D$ by adding or removing a single record (record-level privacy), and for all subsets $O$ of $\mathcal{A}$'s range, $\text{P}(\mathcal{A}(D) \in O) \leq e^\varepsilon \cdot \text{P}(\mathcal{A}(D') \in O) + \delta$. The parameter $\varepsilon \geq 0$ denotes the *privacy budget* or privacy loss, while $\delta \geq 0$ denotes the probability of violation of privacy, with smaller values indicating stronger privacy guarantees in both cases. $\mathcal{A}(D)$ and $\mathcal{A}(D')$ could e.g. be marginal probability distributions estimated on datasets $D$ and $D'$ respectively.

A DP operation $\mathcal{A}$ is commonly created from an operation $f$ by adding noise that is inversely proportional to $\varepsilon$ and proportional to the *sensitivity* of $f$, in which the sen-

sitivity measures the maximum impact a change in the underlying dataset can have on the output of $f$. For example, let $f : \mathbb{D} \to \mathbb{R}^m$ and let $D \sim D'$ denote that $D$ and $D'$ are neighboring datasets, then the $L_2$ sensitivity of $f$ is $\Delta(f) = \max_{D \sim D'} \|f(D) - f(D')\|_2$, while the $L_1$ sensitivity of $f$ is based on the L1-norm, i.e., $\Delta(f) = \max_{D \sim D'} \|f(D) - f(D')\|_1$.

**Definition 3.1** (Gaussian Mechanism). Let $f : \mathbb{D} \to \mathbb{R}^m$, then the Gaussian Mechanism adds i.i.d. Gaussian noise with scale proportional to $\Delta(f)$ to each entry of $f(D)$. Mathematically, $\mathcal{A}(D) = f(D) + \sigma \Delta(f) \mathcal{N}(0, \mathbf{I})$, where $\sigma$ is a scaling coefficient and $\mathbf{I}$ is the $m \times m$ identity matrix.

Similarly, the exponential mechanism is commonly used to make a private choice from a set $\mathcal{R}$ of possible outcomes.

**Definition 3.2** (Exponential Mechanism). Given a quality score function $s : (\mathcal{R}, \mathbb{D}) \to \mathbb{R}$ and a privacy budget $\varepsilon \geq 0$, the exponential mechanism samples an output $r \in \mathcal{R}$ from the probability distribution: $\Pr[\mathcal{A}(D) = r] \propto \exp\left(\frac{\varepsilon \cdot s(r, D)}{2\Delta}\right)$, where $\Delta = \max_{r \in \mathcal{R}} \Delta(s(r, .))$ denotes the global sensitivity of the quality function.

We employ the Gumbel-Max trick as a computationally efficient alternative to the standard exponential mechanism (McSherry & Talwar, 2007). For each candidate $r \in \mathcal{R}$, we compute $s(r, D) + G_r$ where $G_r \sim \text{Gumbel}(0, \beta)$ with scale $\beta = \frac{2\Delta}{\varepsilon}$, and return $\arg\max_{r \in \mathcal{R}}(s(r, D) + G_r)$. This procedure is equivalent to the Report Noisy Max (RNM) algorithm and generates samples from the exact exponential mechanism distribution.

The *post-processing property* of DP guarantees that if $\mathcal{A}$ is DP, then $g \circ \mathcal{A}$ is also DP, where $g$ is an arbitrary function, i.e., any arbitrary computations performed on DP output preserve DP without any effect on the privacy budget $\varepsilon$.

### 3.3. Synthetic Data Generation

A private tabular dataset $D$ consists of $N$ instances, where each instance $x \in D$ is defined by a set of $d$ attributes. Each attribute $x_i$ takes values from a finite, discrete domain $\Omega_i$ of size $|\Omega_i| = \omega_i$. The complete domain of the data is the cartesian product of these individual domains, denoted by $\Omega = \prod_{i=1}^{d} \Omega_i$. Our objective is to generate a synthetic dataset $\widehat{D}$ that accurately reflects the statistical properties of $D$ while satisfying input and output privacy guarantees.

For a subset of attribute indices $w \subseteq \{1, \ldots, d\}$, a marginal $q_w(D)$ is a vector in $\mathbb{R}^{|\Omega_w|}$ representing the frequency in $D$ of each element of the sub-domain $\Omega_w = \prod_{i \in w} \Omega_i$. Specifically, the entry corresponding to a configuration $y \in \Omega_w$ is given by: $q_w(D)_y = \sum_{x \in D} \mathbb{I}(x_w = y)$ where $\mathbb{I}(\cdot)$ is the indicator function and $x_w$ is the projection of $x$ onto the attributes in $w$. A $k$-way marginal refers to a marginal where the subset of attributes has cardinality $|w| = k$.

To generate $\widehat{D}$, we use the marginal-based method AIM (McKenna et al., 2022). It accepts a workload $\mathcal{W} = \{w_1, w_2, \ldots, w_m\}$, which is a pre-defined collection of attribute subsets of interest. Given a total privacy budget $\varepsilon$, the goal is to ensure $q_w(\widehat{D}) \approx q_w(D)$ for all $w \in \mathcal{W}$. To achieve this, AIM first computes noisy 1-way marginals on $D$ to initialize the probabilistic graphical model (**initialization** step, which is a data-dependent operation) and then iteratively constructs the synthetic data distribution by repeating:

1. **Select**: In each iteration $t$, AIM identifies a workload query $w^* \in \mathcal{W}$ that maximizes the error between $D$ and $\widehat{D}$ using the exponential mechanism. The quality score $s^t(w, D)$ for $w$ is given as: $s^t(w, D) = \alpha_w \cdot (\|q_w(D) - q_w(\widehat{D})\|_p - \rho)$ where $\| \cdot \|_p$ denotes a norm (typically the $L_1$ or $L_2$ norm), $\alpha_w$ is a query-specific weight, and $\rho$ is a penalty term. To implement this efficiently, we employ the Gumbel-Max trick: $w^* = \arg\max_{w \in \mathcal{W}}(s^t(w, D) + G_w)$ where $G_w \sim \text{Gumbel}(0, \beta)$ with $\beta$ proportional to the sensitivity of $s^t(w, D)$. For details, please refer to Eq 1 in (McKenna et al., 2022). This step is data-dependent, as the quality score is computed directly using the private database $D$.

2. **Measure**: Once $w^*$ is selected, the algorithm computes a noisy measurement of the true marginal using the Gaussian mechanism $y_{w^*} = q_{w^*}(D) + \mathcal{N}(0, \sigma_t^2 \mathbf{I})$ where the noise scale $\sigma_t$ for the $t^{th}$ iteration is determined by the remaining privacy budget (Note that sensitivity of $q$ is 1). This is a data-dependent step.

3. **Generate**: AIM maintains a compact representation of the synthetic data, typically using a Probabilistic Graphical Model (McKenna et al., 2019). After each measurement, the model is updated to maintain consistency with all previously measured marginals while minimizing the objective function that finds a distribution that best explains the noisy measurements $y_{w^*}$, effectively refining $\widehat{D}$. This step depends only on the noisy measurements and not on the raw data.

*Compute:* In practical implementations of AIM, it is common to first compute the exact values of all marginals in $\mathcal{W}$ on $D$, including the 1-way marginals used for initialization and all higher-order marginals that may be selected during the adaptive iterations. We refer to this data-dependent preprocessing as the **compute** step.[3] This step does not modify the AIM algorithm or its privacy guarantees, since no true marginals are revealed. We note that such precomputation is particularly efficient in FHE (see Prot. 1).

*Privacy Guarantee* (McKenna et al., 2022): By using the Gaussian Mechanism for measurements and the Exponen-

---

[3]In AIM, the precomputed exact answers are used by the select step to evaluate approximation error, while noisy versions are generated only when a marginal is to be measured.

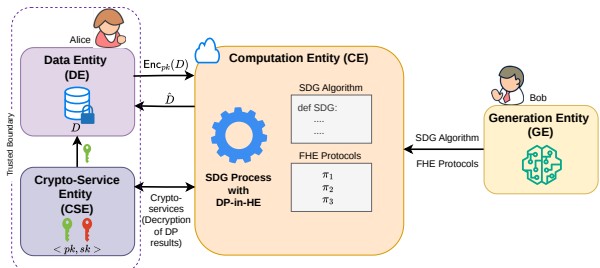

*Figure 1.* System entities (see Sec. 4)

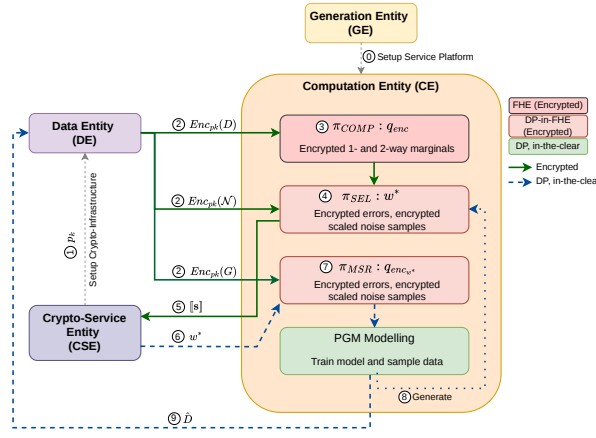

*Figure 2.* FHAIM workflow (see Sec. 4)

tial Mechanism for selection, AIM satisfies $(\varepsilon, \delta)$-DP with respect to $D$. Specifically, the privacy budget is managed using the composition properties of Zero-Concentrated Differential Privacy (zCDP), which are subsequently converted to $(\varepsilon, \delta)$.

## 4. FHAIM: System Description

Let *Alice* be a data holder with a private dataset $D$ and let Bob be a SDG-as-a-service provider who owns an SDG process $\mathcal{G}$. Alice wishes to utilize Bob's service to generate a synthetic dataset $\widehat{D}$ without sharing her raw private data $D$. We propose a system comprising four entities (see Fig. 1), following a model similar to (Ushiyama et al., 2021).

- *Data Entity (DE)*: represents the data holder (*Alice*). The DE encrypts the private dataset $D$ and other auxiliary information (such as generated unit noise samples) using the public key $pk$ received from the Crypto-Service Entity and sends the encrypted data to the Computation Entity. After this initial upload, the DE is not involved in further computations. Its goal is to ensure that the CE learns only $(\epsilon, \delta)$-DP outputs and never observes raw data or unperturbed statistics.
- *Computation Entity (CE)*: provides the computational infrastructure (e.g., Amazon SageMaker) to execute $\mathcal{G}$ and the FHE-based SDG protocols (see Sec. 5). It operates exclusively on encrypted or DP-protected data. Its goal is to execute intensive SDG protocols via encrypted operations and plaintext computations on DP-sanitized outputs, without access to decryption keys or raw plaintext data, strictly under a semi-honest threat model.
- *Generation Entity (GE)*: represents the service provider (*Bob*) who owns the proprietary SDG algorithm $\mathcal{G}$. GE uploads the FHE protocols and SDG logic to the CE. This logical separation allows specialized SDG providers (e.g. MDClone) to outsource FHE execution to cloud platforms (e.g. AWS), enabling flexible, vendor-agnostic deployment. Its goal is to deploy $\mathcal{G}$ without exposing its intellectual property to the client (DE) or assuming legal liability for raw plaintext.
- *Crypto-Service Entity (CSE)*: a trusted party (e.g., Al-

ice or AWS Cryptography) that manages FHE key pairs $\langle pk, sk \rangle$. It provides $pk$ to the DE and uses $sk$ to decrypt *specific* noise-perturbed ciphertexts requested by the CE. Its goal is to act as a restricted decryption oracle within the DE's trust boundary, decrypting only legitimate, DP-sanitized ciphertexts to prevent unperturbed data leakage to the CE.

The above proposed system can be adapted for any SDG algorithm that computes data-dependent statistics. In particular, we adapt the system in Fig. 1 to AIM (McKenna et al., 2022) and develop novel FHE-based SDG protocols to this end. $\pi_{\mathsf{COMP}}$ and $\pi_{\mathsf{MEASURE}}$ are generic for tabular marginals-based algorithms; $\pi_{\mathsf{SELECT}}$ is AIM-specific.

**FHAIM Workflow.** The proposed FHAIM workflow, illustrated in Fig. 2, begins with setting up the service platform and crypto-infrastructure. The CSE generates an FHE key pair and provides the public encryption key $pk$ to the DE (Step ①). The DE then preprocesses and encrypts their private dataset $D$, along with a collection of pre-generated unit noise samples (see Appendix B), and uploads the resulting ciphertexts $Enc_{pk}(D)$, $Enc_{pk}(\mathcal{N})$, and $Enc_{pk}(G)$ to the CE (Step ②). Using the FHE-compatible protocols and iterative logic provided by the GE (in Step ⓪), the CE executes the DP-in-FHE synthetic data generation process. The CE first computes encrypted 1- and 2-way marginals ($q_{enc}$) by executing the FHE protocol $\pi_{\mathsf{COMP}}$ on the encrypted data (Step ③). It then iteratively follows the AIM workflow by executing the FHE protocols $\pi_{\mathsf{SELECT}}$ (Step ④) and $\pi_{\mathsf{MEASURE}}$ (Step ⑦) to obtain an encrypted differentially private marginal $q_{enc_{w^*}}$ for $w^*$. Specifically, as part of $\pi_{\mathsf{SELECT}}$, CE computes and sends the encrypted noisy scores ($[\![\mathbf{s}]\!]$) for all $w$ to the CSE (Step ⑤). The CSE, which is within the trusted boundary of DE, decrypts the perturbed scores, performs the argmax, and returns the plaintext index of the selected marginal $w^*$ to CE (Step ⑥),

ensuring CE only ever observes DP-protected values in-the-clear. CE then uses $w^*$ to complete $\pi_{\text{MEASURE}}$ and obtain $q_{enc_{w^*}}$ (Step ⑦). The CE then executes the modeling in the "generate" step of AIM in the clear (Step ⑧), leveraging the post-processing property of differential privacy to refine the synthetic data distribution without further compromising input privacy. After a fixed number of iterations, the trained DP model finally generates $\widehat{D}$ and sends it to DE (Step ⑨).

**Trust Model.** We consider CE to be semi-honest, i.e., it follows the protocol but attempts to learn about $D$. GE trusts CE with its trademark algorithm. DE does not trust GE. The CSE resides entirely within DE's trust boundary; in practice, the client can act as both DE and CSE, retaining sole control over the secret key.[4] We additionally assume CE and CSE do not collude, i.e. the client does not disclose their secret key to CE.[5]

**Security and Privacy.** FHAIM provides input privacy via FHE and output privacy via DP. CE's view is strictly limited to ciphertexts and DP-protected outputs. The system is designed to provide $(\epsilon, \delta)$-DP guarantees to individuals in $D$. Formally, FHAIM operates in an IND-CPA-D setting, since CE receives decrypted marginals from CSE during the measurement step. We explicitly acknowledge that the CKKS scheme possesses specific vulnerabilities in this decryption-feedback setting, as demonstrated by (Li & Micciancio, 2021). Consequently, while DP bounds the information leakage about the underlying data, securing the FHE scheme against key-recovery attacks in real-world deployments requires independent cryptographic countermeasures, such as careful parameterization or cryptographic noise flooding. A full security analysis is provided in Appendix G.

**Noise Generation.** A central challenge in DP-in-FHE is generating the randomness required to sample DP noise. We adopt a pre-sampling strategy where DE encrypts and uploads a sufficient collection of unit noise samples during the initial upload phase to support the full AIM execution. While this adds a one-time computational burden on the DE, we discuss the challenges and propose strategy in App. B.

## 5. FHAIM: FHE-SDG Protocols

**FHE Protocol for Compute ($\pi_{\text{COMP}}$).** To enable efficient arithmetic, $D$ is first one-hot encoded (OHE), transforming

---

[4]While FHAIM defines four logical roles, the underlying cryptographic construction is a 2-party Client–Server model, where the client (DE + CSE) interacts with the untrusted server (CE + GE), thus serving the notion of a single service provider for a single data holder.

[5]Unlike MPC, where non-collusion is a structural cryptographic requirement across multiple compute servers, here it simply reduces to the client (DE + CSE) not disclosing their secret key to server (CE + GE).

---

categorical attributes into binary vectors. This encoding allows the CE to compute marginal histograms using native additions and multiplications, thereby avoiding costly homomorphic comparison operations. Protocol 1 executes the compute step in FHE. It takes the column-wise encrypted one hot encoded dataset $[\![D]\!]$ to output encrypted 1- and 2-way marginals $[\![q]\!]$ (as is considered by default in AIM implementation).

---

**Protocol 1** $\pi_{\text{COMP}}$: FHE Protocol for **COMPUTE**

**Input:** Encrypted one-hot encoded data $[\![D]\!] \in \{0,1\}^{N \times \sum_{i=1}^{d} \omega_i}$, workload $\mathcal{W}$ with attribute subsets from $D$
**Output:** $[\![q]\!], \forall w \in \mathcal{W}$
1: **for all** $w \in \mathcal{W}$ **do**
2:   **if** $|w| = 1$ **then**
3:     **for** $j \leftarrow 1$ **to** $\omega_w$ **do**
4:       $[\![v_j]\!] \leftarrow \sum_{i=1}^{N} [\![D_w[j]]\!]$
5:     **end for**
6:     $[\![q_w]\!] \leftarrow \textbf{COMBINE}([\![v_1]\!], \dots, [\![v_{\omega_w}]\!])$
7:   **else if** $|w| = 2$, i.e., $w = \{a_1, a_2\}$ **then**
8:     **for** $j \leftarrow 1$ **to** $\omega_{a_1}$ **do**
9:       **for** $k \leftarrow 1$ **to** $\omega_{a_2}$ **do**
10:         $idx \leftarrow j \cdot |\Omega_{a_2}| + k$
11:         $[\![v_{idx}]\!] \leftarrow \sum_{i=1}^{N} [\![D_{a_1}[i,j]]\!] \cdot [\![D_{a_2}[i,k]]\!]$
12:       **end for**
13:     **end for**
14:     $[\![q_w]\!] \leftarrow \textbf{COMBINE}([\![v_1]\!], \dots, [\![v_{\omega_w}]\!])$
15:   **end if**
16: **end for**
17: **return** $[\![q]\!], \forall w \in \mathcal{W}$

---

A primary challenge in this step is determining an efficient encrypted memory layout for the sparse OHE data. A naive design that packs the entire $N \times \sum \omega_i$ matrix into a single ciphertext is infeasible for three key reasons: (i) *Capacity constraints*, as the total data size typically exceeds the slot limits of standard polynomial parameters; (ii) *Alignment costs*, as computing 2-way marginals would necessitate expensive homomorphic rotations to align columns corresponding to different attributes; and (iii) *Ragged domains*, as varying attribute domain sizes $|\Omega_w|$ make uniform packing inefficient.

To overcome these limitations, we propose a *column-wise SIMD packing* strategy. Each binary column of the OHE matrix is encrypted into a distinct ciphertext, utilizing SIMD slots to process up to $L(> N)$ records simultaneously. This design ensures that columns are naturally pre-aligned for interaction, effectively reducing the computational complexity by a factor of $L$. Consequently, computing a 1-way marginal reduces to summing the encrypted column for each bin (requiring $O(\log L)$ rotations at multiplicative depth 0). Similarly, 2-way marginals for attribute pair over $a_1, a_2$, with domain size $\omega_{a_1}, \omega_{a_2}$, require only element-wise multiplication of the respective ciphertexts followed by summation (multiplicative depth 1), eliminating the need for rotation operations during the multiplication phase. See

App. A for illustrative explanations. This strategy ensures strictly $O(N/L)$ linear scaling with respect to the number of records. The overall compute step scales as $O(d^2 \cdot \bar{\omega}^2 \cdot N/L)$, while the 2-way compute cost specifically scales as $\sum_{i<j} \omega_i \omega_j \approx O((\sum_i \omega_i)^2)$, making it quadratic in the total number of OHE columns. From an FHE perspective, this approach is highly scalable: the multiplicative depth depends only on the marginal degree $(k-1)$, not on the dataset size $N$ or domain size $\Omega$. The resulting per-bin scalar ciphertexts are subsequently consolidated into a single packed ciphertext using the COMBINE sub-protocol (see Prot. 4 in App A), which employs a selector plaintext to arrange each scalar into its designated slot for the downstream Select step.

**FHE Protocol for SELECT ($\pi_{\mathsf{SELECT}}$).** The select step (Prot. 2) identifies the workload query $w^*$ that maximizes the error between the real and synthetic data distributions.

For each query $w \in \mathcal{W}$, CE computes an encrypted error vector $[\![d_j]\!]$ by subtracting the estimated answers $[\![q_w(\hat{D})]\!]$ from encrypted true answers $[\![q_w(D)]\!]$ on Line 5 of Prot. 2.

The core challenge in *Select* step lies in defining a numerically stable quality score $s(w, D)$ within the encrypted domain. Standard AIM implementations rely on the $L_1$ norm, but implementing this in FHE presents critical hazards due to the need to approximate the non-polynomial absolute value function. Polynomial approximations face a strict trade-off: low-degree versions degrade utility, while high-accuracy solutions (e.g., the degree-1024 model from the FHERMA challenge[6]) are too computationally expensive for iterative use. Furthermore, these approximations are only stable within fixed ranges (typically $[-1, 1]$), creating a risk of divergence or decryption failure when processing dynamic, data-dependent error magnitudes that exceed these bounds.

To ensure operational robustness, we propose to use the *squared $L_2$ norm* instead. This formulation replaces the approximate absolute value with homomorphic squaring – a native CKKS operation consuming only a single level of multiplicative depth. This eliminates both the accuracy-efficiency tradeoff and the range sensitivity issues, providing a deterministic protocol that is unconditionally stable for arbitrary error magnitudes. We propose the corresponding quality score as below :

$$s^t(w, D) = \alpha_w \left( \left\| q_w(D) - q_w(\hat{D}_{t-1}) \right\|_2^2 - \sigma_t^2 \omega_w \right)$$

Here, $\alpha_w = \sum_{x \in \mathcal{W}} c_x \cdot |w \cap x|$, where $c_x$ is the weight assigned to the workload query $x$. The penalty term $\rho = \sigma^2 \omega_w$ (see Theorem D.2) and the sensitivity $\Delta s(w, D) = \max_{w \in \mathcal{W}} \alpha_w (2N + 1)$ (see Theorem D.1).

[6]https://fherma.io/content/65de3f45bfa5f4 ea4471701c

---

**Protocol 2 $\pi_{\mathsf{SELECT}}$: FHE Protocol for SELECT**

**Input:** Candidate workload queries $Q_C \subseteq \mathcal{W}$, encrypted marginals $[\![q]\!]$, estimated marginals $\hat{q}$, privacy parameter $\varepsilon$, sensitivity $\Delta$, bias vector $\mathbf{b}$, weights vector $\mathbf{wt}$
**Output:** Selected query $w^*$
1: Initialize $[\![\mathbf{s}]\!]$ of length $|Q_C|$ {vector of quality scores}
2: **for all** $w \in Q_C$ **do**
3:    $[\![\hat{q}_w]\!] \leftarrow \mathrm{ENC}(\hat{q}_w)$ {encrypt estimates}
4:    **for** $j \leftarrow 1$ to $\omega_w$ **do**
5:      $[\![d_j]\!] \leftarrow [\![q_w[j]]\!] - [\![\hat{q}_w[j]]\!]$
6:    **end for**
7:    $[\![\mathbf{d}_w]\!] \leftarrow \mathrm{COMBINE}([\![d_1]\!], \ldots, [\![d_{\omega_w}]\!])$
8:    $[\![s_w]\!] \leftarrow \mathrm{SQUAREDNORM}([\![\mathbf{d}_w]\!])$
9:    $[\![s_w]\!] \leftarrow ([\![s_w]\!] - b_w) \cdot wt_w$
10:    *// Add Gumbel Noise (Encrypted Gumbel-Max)*
11:    $[\![s_w]\!] \leftarrow [\![s_w]\!] + [\![G_w]\!] \cdot \frac{2\Delta}{\epsilon}$
12:    $[\![\mathbf{s}[w]]\!] \leftarrow [\![s_w]\!]$
13: **end for**
14: CE sends $[\![\mathbf{s}]\!]$ to CSE
15: *// CSE decrypts and performs argmax in the clear*
16: $w^* \leftarrow \arg\max_{w \in Q_C} \mathrm{DEC}([\![\mathbf{s}]\!])$
17: **return** $w^*$ {CSE sends $w^*$ to CE}

---

In $\pi_{\mathsf{SELECT}}$, the CE computes the encrypted quality scores following the above formulation on Lines $7 - 9$. Empirical results confirm that while the $L_1$ implementation suffers from utility loss due to approximation errors, the squared $L_2$ approach produces results nearly identical to the plaintext baseline (see Sec. 6 and Tables 3 and 5 in App. F).

To compute the $w^*$ (see select step in Sec. 3.3), the CE scales the pre-encrypted Gumbel noise by $\frac{2 \cdot \Delta}{\varepsilon}$ where $\Delta = \max_{w \in \mathcal{W}} \Delta(s(w, \cdot))$, and adds the encrypted scaled noise directly to the encrypted quality scores on Line 11. The CE then sends the encrypted noisy scores $[\![\mathbf{s}]\!]$ for all $w \in Q_C$ to the CSE on Line 14. The CSE, being a trusted entity within the privacy boundary, decrypts the vector to find the argmax in the clear (on Line 16). The CSE sends $w^*$ to CE on Line 17, ensuring that the CE only receives the DP selected index, maintaining the standard $(\epsilon, \delta)$-DP guarantee of the Report Noisy Max mechanism.

The role of CSE in $\pi_{\mathsf{SELECT}}$ can be minimized at the cost of additional cryptographic overhead. FHE protocols for argmax, such as those based on polynomial approximation (e.g. (Zhang et al., 2024)) or scheme-switching, could in principle be implemented and integrated into FHAIM, thus avoiding the computations by CSE. However both these methods would incur both computational and memory cost because of either increased multiplicative depth requirement (in chained polynomial evaluation) or latency of scheme-switching itself. For example, in our experiments on the COMPAS dataset with scheme-switching implementation of argmax, the latency increased by 6x-7x (See Table in 4 in App. F). Adopting such FHE protocols, the role of the

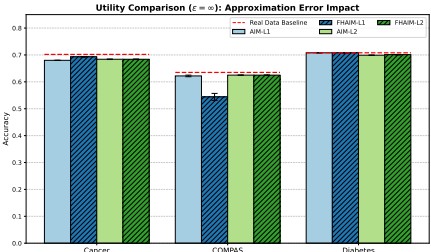
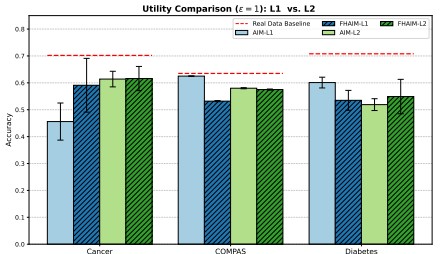

*(a)* $\epsilon = \infty$ **(No DP Noise).** FHAIM-L1 suffers accuracy loss (e.g., COMPAS) purely due to approximation errors, while FHAIM-L2 remains stable.

*(b)* $\epsilon = 1$ **(High Privacy).** FHAIM-L2 (hatched green) matches the AIM-L2 baseline, whereas FHAIM-L1 degrades significantly due to instability.

*Figure 3.* **Utility Comparison.** Classification accuracy across three datasets. The results isolate the impact of polynomial approximation errors in the $L_1$ norm (blue bars) versus the stability of the exact squared $L_2$ norm (green bars). Red dashed lines indicate real data performance.

CSE can instead be minimized to a simple decryption step for determining the selected index albeit at a substantial performance cost. In our implementation, we prioritize correctness-under-trust over costly cryptographic operations. Consequently, under this efficient design, the computational cost of the select step scales polynomially with the number of attributes as $O(d^3)$.

**FHE Protocol ($\pi_{\text{MEASURE}}$).** The measure step (See Protocol 3 and details in the App. A) computes a noisy marginal for a given workload $w^*$ by adding encrypted Gaussian noise to the encrypted marginal answer, implementing the Gaussian mechanism entirely within the encrypted domain.

## 6. Results

We evaluate FHAIM on three datasets: breast-cancer (Zwitter & Soklic, 1988) with 285 patients and 10 attributes, prison recidivism (COMPAS)[7] (Angwin et al., 2016) with 7,214 individuals and 7 attributes, and diabetes (Smith et al., 1988) with 768 patients and 9 attributes. Each dataset is randomly split into train and test subsets using a 80%/20% split ratio. More details on the datasets are in Appendix E.1.

We implemented FHAIM using OpenFHE with the CKKS-RNS scheme (Cheon et al., 2017), operating as leveled FHE without bootstrapping. For 128-bit security, we set ring dimension $N = 2^{15}$, batch size 16,384, scaling modulus 59 bits, and first modulus 60 bits (total modulus 650 bits). Though the theoretical minimum multiplicative depth is low ($\approx 2$ for FHAIM-L2, $\approx 5$ for FHAIM-L1), we set it to 10. This provides a precision buffer for CKKS approximate arithmetic and accommodates extra levels consumed by OpenFHE's `EvalPoly` implementation.

---

[7]https://www.propublica.org/datastore/dataset/compas-recidivism-risk-score-data-and-analysis

### 6.1. Utility

In Figure 3b and Figure 3a , we report the statistical utility of the generated data as the workload error ($\Delta$): $\Delta(D, \widehat{D}) = \frac{1}{|\mathcal{W}|} \sum_{w \in \mathcal{W}} \|q_w(D) - q_w(\widehat{D})\|$. We also report the utility of the generated data by training a logistic regression model on the synthetic data and evaluating the trained models on the test data. Figures 3a and 3b compare three experimental settings: the real data baseline (red dashed lines), the non-private $\epsilon = \infty$ setting (Fig. 3a), and the strict $\epsilon = 1$ privacy setting (Fig. 3b) . The results validate the privacy-utility trade-off: model fidelity is highest on real data and exhibits a natural, expected reduction under strict privacy constraints ($\epsilon = 1$), reflecting the unavoidable cost of rigorous differential privacy. The $\epsilon = \infty$ results (Fig. 3a) confirm that without noise, the underlying synthesis logic is sound, while the $\epsilon = 1$ case (Fig. 3b) demonstrates that FHAIM-L2 maintains this stability even under high-privacy conditions.

The $\varepsilon = \infty$ results (Figure 3a) isolate structural fidelity, revealing a significant utility gap where FHAIM-L1 underperforms the plaintext baseline due to accumulated errors from approximating the non-polynomial $L_1$ norm . Conversely, FHAIM-L2 eliminates this gap by leveraging the exact homomorphic squaring of the $L_2$ norm, achieving parity with the plaintext model . This robustness persists under strict privacy constraints ($\varepsilon = 1$, Figure 3b), confirming that FHAIM-L2 preserves utility while providing input privacy, with only minor stochastic variations attributable to DP noise.

### 6.2. Computation Cost

We evaluate the efficiency of FHAIM by analyzing the runtime and memory footprint across the three phases of the protocol: COMPUTE, SELECT, and MEASURE on a 16-Core CPU and 64 GB RAM running Ubuntu 24.04 LTS. Experiments were conducted on a standard cloud instance. Figure 4 illustrates the runtime breakdown, and Table 2 details peak

memory usage.

**Runtime Analysis: The Amortization of Privacy.** As illustrated in Figure 4, the computational cost is heavily skewed toward the one-time initialization phase, allowing the iterative steps to proceed efficiently.

- **Initialization (Compute Step):** The COMPUTE phase is the most computationally intensive, ranging from approximately 5 minutes (COMPAS) to 12 minutes (Cancer). Crucially, this is a *one-time cost*. Because the generation of 1- and 2-way marginals depends only on the raw data and the workload $\mathcal{W}$, it is executed once prior to the iterative training loop. As expected, the runtime is identical for both L1 and L2 variants, as the choice of error metric does not influence the initial marginal computation.
- **Iterative Training (Select & Measure):** The training loop is dominated by the SELECT step. The FHAIM-L2 protocol incurs a modest computational overhead compared to FHAIM-L1 (e.g., $\sim 15\%$ increase for Cancer). While the Squared $L_2$ norm requires fewer multiplicative levels (depth 1) than the polynomial approximation of $L_1$ (depth 4 for degree-10), the operations involve larger ciphertext scales to preserve precision without rescaling, leading to slightly longer execution times. However, this trade-off is negligible in practice: an average iteration takes $\sim 1 - 2$ minutes. For a typical run of $T = 16d$ iterations, the total training time remains within the manageable range of 15 to 45 minutes for all datasets tested.
- **The Measure Step:** The MEASURE step is extremely efficient, executing in under 2.5 seconds per iteration across all datasets. Its contribution to the total runtime is effectively negligible.

**Memory Footprint.** Table 2 reports the peak memory usage during execution. FHAIM demonstrates a remarkably low memory footprint, with peak usage remaining under **32 MB** for all datasets. While FHAIM-L2 generally consumes more memory than FHAIM-L1 (e.g., 31.72 MB vs. 11.12 MB for COMPAS), this increase is an artifact of the ciphertext maintenance required for the squared norm operations. Nevertheless, these requirements are orders of magnitude lower than the available RAM on standard commercial hardware, confirming that FHAIM can be deployed on lightweight computing instances without memory bottlenecks.

*Table 2.* Peak Memory Usage (MB)

| Dataset | Dataset Size ($N \times d$) | FHAIM-L1 | FHAIM-L2 |
|---|---|---|---|
| Breast Cancer | $228 \times 10$ | 18.95 | 26.28 |
| COMPAS | $4120 \times 7$ | 11.12 | 31.72 |
| Diabetes | $614 \times 9$ | 13.72 | 26.57 |

**Complexity and Scalability.** The use of SIMD packing ensures that our runtime scales efficiently with dataset size

$N$. As discussed in Section 5, the multiplicative depth of our circuit depends only on the marginal degree $k$, not $N$. Consequently, increasing the number of records primarily affects the number of ciphertexts processed in parallel, which scales linearly $O(N/L)$ rather than super-linearly. This confirms that FHAIM is scalable to larger datasets without requiring exponential increases in compute resources.

Furthermore, to evaluate scalability on larger benchmarks, we tested FHAIM on the **Adult dataset** (Kohavi et al., 1996) used in the original AIM paper. While the plaintext AIM algorithm completed generation in 19 minutes (AIM-L1) and 5 minutes (AIM-L2), FHAIM required approximately 5.5 hours and 5.3 hours, respectively. This increased runtime reflects the computational cost of scaling the FHE parameters; specifically, accommodating the larger number of records required increasing the ring dimension to $N = 2^{17}$. Since SDG is not a real-time sensitive task, we believe these runtimes are a tolerable one-time cost, given the benefits of generating synthetic data while preserving both input and output privacy.

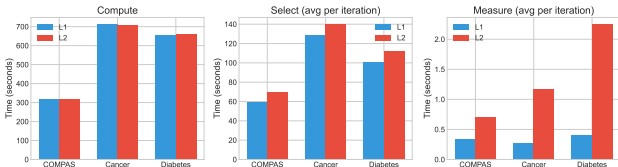

*Figure 4.* **Runtime Analysis.** Breakdown of execution times (in seconds) for the Compute, Select, and Measure protocols across datasets. The one-time **Compute** phase (left) dominates the total runtime and is invariant to the choice of norm. In the iterative **Select** (center) and **Measure** (right) phases, the exact Squared $L_2$ protocol (red) incurs a modest computational overhead compared to the degree-10 $L_1$ approximation (blue), reflecting the trade-off between the raw speed of low-degree approximations and the numerical stability of the exact squared norm.

## 7. Conclusion

We introduced FHAIM, the first framework enabling input-private synthetic data generation (SDG) by training tabular SDG models directly over fully homomorphically encrypted data. By combining novel FHE protocols for marginal computation, DP noise injection, and encrypted query selection, FHAIM demonstrates that both input and output privacy can be achieved within an efficient SDG-as-a-service workflow. Our experiments show that FHAIM preserves the statistical utility and downstream machine learning (ML) performance of the marginal-based AIM SDG algorithm, especially under the more stable squared-$L_2$ objective, while maintaining practical runtimes on real datasets. These results establish FHE-based SDG as a viable and scalable primitive for privacy-preserving data sharing, and open new avenues for secure outsourcing of ML workflows where both data confidentiality and formal privacy guarantees are essential.

## Impact Statement

This work aims to advance privacy-preserving machine learning by enabling synthetic data generation directly on encrypted data. The primary societal benefit is facilitating the safe sharing of sensitive information in critical domains – such as healthcare and finance – without requiring data holders to trust a central server. We acknowledge that the use of Fully Homomorphic Encryption introduces a significant computational overhead compared to plaintext approaches, resulting in a higher energy footprint. We do not foresee other immediate negative societal consequences beyond these resource considerations.

## Acknowledgements

This material is based upon work supported by the National Science Foundation under Grant Nos. 2451163, 2523406, CCF-2523407, and CNS-2413232, and by NSF NAIRR 240485 (Cloudbank AWS) and NSF NAIRR 240091 (TACC Frontera). This research was, in part, funded by the National Institutes of Health (NIH) Agreement No. 1OT2OD032581. The views and conclusions contained in this document are those of the authors and should not be interpreted as representing the official policies, either expressed or implied, of the NIH. This work is supported by eScience Institute.

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

# A. FHE primitives

Homomorphic encryption is a cryptographic method that enables data processing while the information remains in an encrypted state. Because the raw data is never exposed during computation, this technique is essential for secure cloud computing and third-party data processing.

Formally, the scheme consists of a tuple of four probabilistic polynomial-time algorithms $\Pi_{\text{HE}} = (\text{KeyGen}, \text{Enc}, \text{Eval}, \text{Dec})$ defined as follows:

- HE.KeyGen$(\lambda)$: Given the security parameter $\lambda$ (which determines the polynomial degree $N$ and modulus $Q$), the key generation algorithm outputs a key pair $(pk, sk)$ and the related cryptographic context. Crucially for our FHAIM implementation, this context includes auxiliary *evaluation keys* (such as relinearization and rotation keys) required to perform homomorphic multiplication and SIMD slot rotations efficiently.

- HE.Enc$(pk, m)$: The encryption algorithm takes in the public key $pk$ and a plaintext message $m$, then outputs the ciphertext $c$. In the CKKS scheme used here, $m$ is typically a vector of real numbers encoded into the slots of a polynomial ring. For our purposes, $m$ corresponds to the flattened vectors of one-hot encoded attributes or pre-generated noise samples.

- HE.Eval$(c, f)$: The encrypted evaluation algorithm takes in a ciphertext message $c$ and a function $f$ (represented as an arithmetic circuit), then outputs the computation result $c'$. Here, $f$ encompasses the arithmetic operations required for marginal computation (additions and multiplications) and the squared $L_2$ error metric evaluation.

- HE.Dec$(sk, c')$: The decryption algorithm takes in the secret key $sk$ and a ciphertext message $c'$, then outputs the plaintext $m'$. Due to the approximate arithmetic of CKKS, the recovered message satisfies $m' \approx f(m)$, containing a negligible error term provided the accumulated noise remains within the decryption radius.

- HE.Add$(c_1, c_2)$: The homomorphic addition algorithm takes two ciphertexts $c_1$ and $c_2$ (encrypting $m_1$ and $m_2$) and outputs a ciphertext $c_{\text{add}}$ such that HE.Dec$(sk, c_{\text{add}}) \approx m_1 + m_2$. This operation is computationally inexpensive and consumes negligible multiplicative depth. In Protocol 1, we utilize this extensively to aggregate one-hot encoded counts for 1-way marginals.

- HE.Mult$(c_1, c_2)$: The homomorphic multiplication algorithm outputs a ciphertext $c_{\text{mult}}$ encrypting the element-wise product $m_1 \cdot m_2$. This operation significantly increases the noise level and consumes one level of multiplicative depth. It is typically followed by a *relinearization* step (using evaluation keys) to prevent ciphertext size expansion. We employ this primitive for computing 2-way marginal interactions and for the squaring operation in the $L_2$ quality score calculation.

- HE.Rot$(c, k)$: Given a ciphertext $c$ encrypting a vector $m$, the rotation algorithm outputs a ciphertext $c_{\text{rot}}$ encrypting a cyclically shifted vector $m'$ where $m'[i] = m[(i + k) \pmod{L}]$. This operation requires specific rotation keys generated during HE.KeyGen. In Protocol 2, this primitive is essential for the COMBINE sub-protocol, allowing us to move scalar values into specific SIMD slots to pack error vectors efficiently.

$\pi_{\text{MEASURE}}$: **FHE Protocol for MEASURE** For each bin $j \in \{1, 2, \ldots, |\Omega_w|\}$, the CE scales a pre-encrypted unit Gaussian noise sample $[\![z_j]\!]$ by $\sigma$ and adds it to the encrypted marginal entry $[\![q_w[j]]\!]$. The resulting noised ciphertext is then sent to the CSE for decryption. Since the DP noise is added before decryption, the CSE only ever observes the already-privatized marginal $\hat{q}_{w^*}$, ensuring that the true marginal is never revealed in the clear to any entity.

**FHE Protocol for Combine** ($\pi_{\text{COMB}}$). The combine sub-protocol (Protocol 4) packs $k$ scalar ciphertexts — each holding a value in slot 0 — into a single ciphertext vector where position $i$ holds the $i$-th value. This is done using a pre-computed selector plaintext $\mathbf{s}$ with $\mathbf{s}[0] = 1$ and all other entries 0. For each scalar ciphertext $[v_i]$, the CE multiplies it by $\mathbf{s}$ to isolate the value in slot 0, then applies a homomorphic rotation by $i$ to shift it into position $i$, and accumulates the result. This sub-protocol is used in both $\pi_{\text{COMP}}$ to pack marginal entries and in $\pi_{\text{SELECT}}$ to pack the error vector for norm computation.

---

**Protocol 3** $\pi_{\text{MEASURE}}$: FHE Protocol for **MEASURE**

---

**Input:** Selected query $w^*$, encrypted marginal $\llbracket q_{w^*} \rrbracket$, noise scale $\sigma$, encrypted Gaussian noise buffer $\llbracket z \rrbracket$, counter $c$
**Output:** Noised marginal $\tilde{q}_{w^*}$

1: *//number of attributes in $w$*
2: $m \leftarrow |w^*|$
3: **for** $j \leftarrow 1$ to $\omega_{w^*}$ **do**
4: $\quad \llbracket \tilde{q}_{w^*}[j] \rrbracket \leftarrow \llbracket q_{w^*}[j] \rrbracket + \sigma \cdot \llbracket z[c+j] \rrbracket$
5: $\quad \tilde{q}_{w^*}[j] \leftarrow \text{DEC}(\llbracket \tilde{q}_{w^*}[j] \rrbracket)$
6: **end for**
7: $c \leftarrow c + m$
8: **return** $\tilde{q}_{w^*}$

---

**Protocol 4** $\pi_{\text{COMB}}$: FHE Sub-Protocol for **COMBINE**

---

**Input:** Scalar ciphertexts $\llbracket v_1 \rrbracket, \ldots, \llbracket v_k \rrbracket$, selector plaintext $\mathbf{s}$ with $\mathbf{s}[0] = 1$ and $\mathbf{s}[j] = 0$ for $j > 0$
**Output:** Packed ciphertext $\llbracket \mathbf{v} \rrbracket$ with $\mathbf{v}[i] = v_i$

1: $\llbracket \mathbf{v} \rrbracket \leftarrow \text{ENC}(\mathbf{0})$
2: **for** $i = 0$ to $k - 1$ **do**
3: $\quad \llbracket \mathbf{v} \rrbracket \leftarrow \llbracket \mathbf{v} \rrbracket + \text{ROT}(\mathbf{s} \cdot \llbracket v_i \rrbracket, i)$
4: **end for**
5: **return** $\llbracket \mathbf{v} \rrbracket$

---

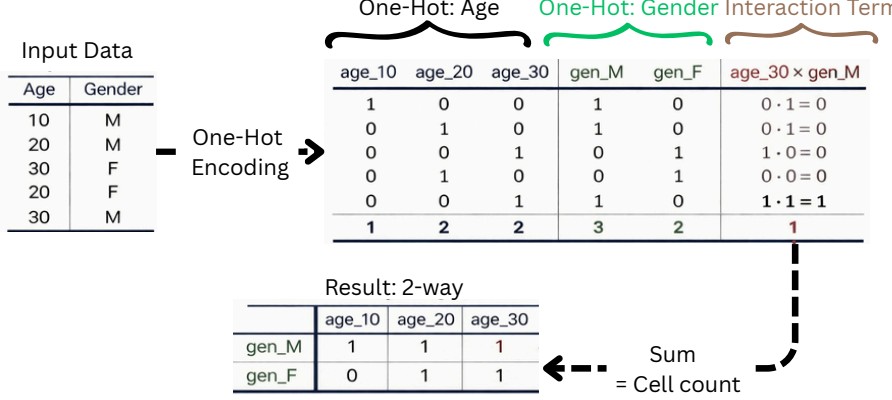

*Figure 5.* Illustration of 2-way marginal computation

**Note on one-hot encoding (OHE)** To compute marginals efficiently in FHE, we one-hot encode (OHE) $D$. Each attribute $x_i$ with domain size $\omega_i$ is transformed into a binary vector of length $\omega_i$ where only the index corresponding to the attribute value is set to 1. In Figure 5, we consider two attributes 'Age' and 'Gender' with $\Omega_{Age} = \{10, 20, 30\}$ and $\Omega_{Gender} = \{M, F\}$. The OHE data then is tranformed to a matrix of size $N \times (|\Omega_{Age}| + |\Omega_{Gender}|)$.

To compute 1-way marginal for 'Age' attribute, we need to add each entry in the OHE'd column (as shown in the last row of top right table) and there are 3 such columns. In general, computing the 1-way marginal for attribute $x_i$ with domain $\omega_i$ requires $\omega_i \cdot (N - 1)$ additions.

Computing a 2-way marginal is illustrated for computing one cell corresponding to age_30, gen_M in the resulting 2-way marginal for 'Age', 'Gender'. The corresponding to age_30, gen_M columns are multiplied resulting in a new column shown by the interaction term and then the values are added up. In general, computing the 2-way marginal for attribute pair $(x_i, x_j)$ requires $\omega_i \cdot \omega_j \cdot N$ multiplications to determine the intersection of one-hot encoded features for each record, followed by $\omega_i \cdot \omega_j \cdot (N - 1)$ additions to aggregate these counts[8]

Generalizing this, computing a $k$-way marginal for a subset of attributes $\mathcal{C}$ requires $N(k-1) \prod_{i \in \mathcal{C}} \omega_i$ multiplications and $(N-1) \prod_{i \in \mathcal{C}} \omega_i$ additions.

---

[8]Using comparisons, we would need $\omega_i \cdot N$ equality checks and $\omega_i \cdot N$ additions for computing a 1-way marginal.

## B. Noise Generation and DP-in-FHE

A central challenge in DP-in-FHE is generating the randomness required to sample DP noise. We adopt a pre-sampling strategy where DE encrypts and uploads a sufficient collection of unit noise samples during the initial upload phase to support the full AIM execution. The CE then homomorphically scales this noise based on the specific privacy budget $\varepsilon$ and the global sensitivity of the queries. To allow the DE to remain offline and to preserve the client–server deployment model, we derive data-independent upper bounds on the number of required noise samples. The total noise requirement follows directly from the two phases of AIM: (i) the initialization phase, which measures all 1-way marginals (Algorithm 2 in (McKenna et al., 2022)), and (ii) the iterative select–measure–generate loop (Algorithm 3 in (McKenna et al., 2022)). Following AIM, we upper bound the number of selection rounds by $T = 16d$.

- *Gaussian Noise Samples (Measure)* During initialization, all 1-way marginals are measured once. Furthermore, in each of the $T$ rounds, a single selected marginal is measured. To obtain a data-independent bound on the amount of noise samples required, we assume that each selected marginal has the maximum possible domain size. The total number of required Gaussian noise samples is therefore

$$\left(\sum_{i=1}^{d} |\omega_i|\right) + \left(16d \cdot \max_{w \in \mathcal{W}} |\omega_w|\right) \tag{1}$$

- *Gumbel Noise Samples (Select)*: In each of the $T$ rounds, DP noise is added to the candidate workload $\mathcal{C} \subseteq \mathcal{W}$ during query selection. We conservatively bound the total number of required Gumbel noise samples by

$$16d \cdot |\mathcal{W}| \tag{2}$$

While this adds a one-time computational burden on the DE, it is a necessary requirement as it is challenging for CE to generate secret randomness within the FHE domain. Note that the CE cannot add its own known plaintext noise, as it then could easily subtract that noise after the CSE decrypts the result, thereby exposing the raw data. By providing encrypted noise, the DE ensures that the statistics are privatized via "DP-in-FHE" before they are ever observed in plaintext by the service provider.

The noise samples are chunked and packed into ciphertexts for efficient summation with marginals as needed.

## C. Polynimial approximation of $|x|$

The L1-norm requires approximating the absolute value function, which has no closed-form polynomial representation. We approximate $|x|$ over $[-1, 1]$ using a degree-10 least-squares polynomial:

$$p(x) \quad = \quad 0.0556 \quad + \quad 3.6049x^2 \quad - \quad 11.9929x^4 \quad + \quad 24.4175x^6 \quad - \quad 23.6236x^8 \quad + \quad 8.5577x^{10} \tag{3}$$

To extend the approximation to an arbitrary range $[-\alpha, \alpha]$, we evaluate $\alpha \cdot p(x/\alpha)$. Value of $\alpha$ used is 10000.

## D. Optimizations for Select Step

We propose the following score function to use squared $L_2$ norm.

$$s^t(w, D) = \alpha_w \left( \left\| q_w(D) - q_w(\hat{D}_{t-1}) \right\|_2^2 - \sigma_t^2 \omega_w \right) \tag{4}$$

**Theorem D.1** (Sensitivity of Squared $L_2$ Quality Score). *Let $q_w(D)$ be a marginal and $\hat{q}_w$ be the estimated marginal from the graphical model. Define the quality score as $s(w,D) = \alpha_w \left( \left\| q_w(D) - q_w(\hat{D}) \right\|_2^2 - \rho \right)$, where Under unbounded differential privacy, the global sensitivity of $s(w, .)$ is $\Delta s(w, .) = |\alpha_w|(2N + 1)$, where $N$ is the number of records, $\rho$ is penalty term and $\alpha_w$ is the weight assigned to $w$.*

*Proof.* Let $D$ and $D'$ be neighboring datasets such that $D' = D \cup \{z\}$, where $z$ is a single record. Adding $z$ increments exactly one bin in the marginal $q_w(D)$. Let $x$ be the count of that bin in $D$, and $\hat{x}$ be the corresponding estimate in $q_w(\hat{D})$.

The change in the quality score is ($\rho$ cancels out):

$$
\begin{aligned}
|s(w,D') - w(w,D)| &= |\alpha_w| \cdot |((x+1) - \hat{x})^2 - (x - \hat{x})^2| \\
&= |\alpha_w| \cdot |(x - \hat{x} + 1)^2 - (x - \hat{x})^2| \\
&= |\alpha_w| \cdot |(x - \hat{x})^2 + 2(x - \hat{x}) + 1 - (x - \hat{x})^2| \\
&= |\alpha_w| \cdot |2(x - \hat{x}) + 1|
\end{aligned}
$$

In the worst case, the error $|x - \hat{x}|$ is bounded by the total number of records $N$. Thus, the global sensitivity is $\Delta s(w, D) = |\alpha_w|(2N + 1)$. $\qquad\square$

To apply the exponential mechanism to select a candidate, we use $\Delta s(w, D) = \max_{w \in \mathcal{W}} \alpha_w(2N + 1)$. The privacy analysis follows zCDP compositions and does not change. The only change here is the sensitivity parameter and not the composition or analysis.

**Theorem D.2** (Expected Squared $L_2$ Penalty). *Let* $\mathbf{y} = q_w(D) + \eta$ *be a noisy measurement where* $\eta \sim \mathcal{N}(0, \sigma^2 \mathbf{I}_{\omega_w})$. *The expected squared $L_2$ error introduced solely by the Gaussian noise is* $\mathbb{E}[\|\eta\|_2^2] = \omega_w \sigma^2$

*Proof.* The squared $L_2$ norm of the noise vector $\eta$ is the sum of the squares of its individual components:

$$
\|\eta\|_2^2 = \sum_{i=1}^{\omega_w} \eta_i^2
$$

By the linearity of expectation, the expected value of a sum is equal to the sum of the expected values:

$$
\mathbb{E}[\|\eta\|_2^2] = \mathbb{E}\left[\sum_{i=1}^{\omega_w} \eta_i^2\right] = \sum_{i=1}^{\omega_w} \mathbb{E}[\eta_i^2]
$$

The expectation of a squared random variable, $\mathbb{E}[X^2]$ is given by:

$$
\begin{aligned}
\mathrm{Var}(X) &= \mathbb{E}[X^2] - (\mathbb{E}[X])^2 \\
\mathbb{E}[X^2] &= \mathrm{Var}(X) + (\mathbb{E}[X])^2
\end{aligned}
$$

For Gaussian noise $\eta_i \sim \mathcal{N}(0, \sigma^2)$, the mean ($\mathbb{E}[\eta_i]$) is 0 and the variance ($\mathrm{Var}(\eta_i)$) is $\sigma^2$. Substituting these values into the identity:

$$
\begin{aligned}
\mathbb{E}[\eta_i^2] &= \sigma^2 + (0)^2 \\
\mathbb{E}[\eta_i^2] &= \sigma^2
\end{aligned}
$$

Summing over all $\omega_w$ bins, we obtain:

$$
\mathbb{E}[\|\eta\|_2^2] = \sum_{i=1}^{\omega_w} \sigma^2 = \omega_w \sigma^2
$$

This constant $\sigma^2 \omega_w$ is the penalty term in the quality score $s(w,D)$ in Equation D. $\qquad\square$

# E. Data, Modelling and Parameters

### E.1. Data

The breast-cancer dataset has 10 categorical attributes and 285 samples. The COMPAS data consists of categorical data. We utilize the same version as in (Calmon et al., 2017), which consists of 7 categorical features and 7,214 samples. The diabetes dataset has 9 continuous attributes and 768 samples.

### E.2. Modelling

To test the utility of the generated data, we train logistic regression models. For breast-cancer, the task is to predict if the cancer will recur. For COMPAS, the task is to predict whether a criminal defendant will re-offend. For the diabetes dataset, the task is to classify a patient as diabetic. We train ML models on real train data and synthetic data independantly. We use a random seed of 42 to train the models. For AIM, we use default parameters, except for $\epsilon$. We note that the original AIM algorithm itself takes too long to run for large datasets (as also reported in (Chen et al., 2025)).

### E.3. Communication Cost

The Data Entity (DE) uploads 45, 24, and 39 ciphertexts ($\approx$ 270MB, $\approx$ 144MB, and $\approx$ 234MB) for the Breast Cancer, COMPAS, and Diabetes datasets respectively, plus a small number of noise ciphertexts. Although One-Hot Encoding (OHE) can make the dataset appear large, our column-wise SIMD packing strategy encodes all records into a single ciphertext per OHE column. Consequently, the upload size scales with the total domain size $\sum \omega_i$ rather than the dataset size $N$. Furthermore, this upload is a strictly one-time cost, after which the DE goes fully offline.

## F. Additional Experiments

*Table 3.* **Utility of Synthetic Data.** AIM-L1 is the original AIM algorithm

| | Method | \multicolumn{3}{c}{CANCER $N = 228, d = 10, \|\Omega\| = 598,752$} | | | \multicolumn{3}{c}{COMPAS $N = 4120, d = 7, \|\Omega\| = 5832$} | | | \multicolumn{3}{c}{DIABETES $N = 614, d = 9, \|\Omega\| = 375,000$} | | |
|---|---|---|---|---|---|---|---|---|---|---|
| | | $\Delta$ | Acc. | F1 | $\Delta$ | Acc. | F1 | $\Delta$ | Acc. | F1 |
| | Real data | – | 0.702 | 0.585 | – | 0.635 | 0.625 | – | 0.708 | 0.602 |
| Synthetic $\epsilon = \infty$ | AIM-L1 | 0.057 | 0.680 | 0.318 | 0.013 | 0.622 | 0.605 | 0.297 | 0.708 | 0.582 |
| | FHAIM-L1 | 0.105 | 0.693 | 0.104 | 0.031 | 0.544 | 0.139 | 0.321 | 0.708 | 0.491 |
| | AIM-L2 | 0.058 | 0.684 | 0.284 | 0.014 | 0.625 | 0.610 | 0.300 | 0.699 | 0.577 |
| | FHAIM-L2 | 0.058 | 0.684 | 0.321 | 0.014 | 0.625 | 0.610 | 0.298 | 0.701 | 0.583 |
| Synthetic $\epsilon = 1$ | AIM-L1 | 0.415 | 0.456 | 0.338 | 0.019 | 0.625 | 0.628 | 0.361 | 0.601 | 0.299 |
| | FHAIM-L1 | 0.475 | 0.591 | 0.349 | 0.035 | 0.532 | 0.504 | 0.382 | 0.535 | 0.355 |
| | AIM-L2 | 0.441 | 0.614 | 0.307 | 0.028 | 0.580 | 0.597 | 0.435 | 0.519 | 0.278 |
| | FHAIM-L2 | 0.476 | 0.616 | 0.495 | 0.028 | 0.575 | 0.620 | 0.444 | 0.549 | 0.218 |

*Table 4.* Latency Impact of FHE Argmax via Scheme-Switching (COMPAS Dataset)

| Argmax Implementation | Execution Environment | Select Step Latency (per iteration) |
|---|---|---|
| Trusted CSE (FHAIM Default) | Cleartext (Post-Decryption) | 60s (Baseline) |
| Scheme-Switching | Encrypted Domain | 390s (6.5$\times$ Increase) |

*Table 5.* **Computational Cost**: `FHAIM` Runtimes (L1/L2)

| | Subprot. | CANCER | COMPAS | DIABETES |
|---|---|---|---|---|
| $\pi_{\text{COMP}}$ | $\pi_{\text{1way}}$ | 31s | 25s | 32s |
| | $\pi_{\text{2way}}$ | 681s | 292s | 625s |
| $\pi_{\text{SELECT}}$ | $\pi_{\text{ERR}}$ | 129/140s | 59/70s | 101/112s |
| | $\pi_{\text{GUMBEL}}$ | 4.8s | 3.9s | 4.0s |
| $\pi_{\text{MEASURE}}$ | $\pi_{\text{GAUSS}}$ | 0.28/1.18s | 0.34/0.70s | 0.40/2.25s |

## G. Security Proof

Let $\{\tilde{q}_j\}_{j=1}^{M}$ denote the collection of all $M = d + T$ noisy marginals returned to the CE: the $d$ noisy 1-way marginals from initialization and the $T$ noisy measured marginals from the iterative loop. Each is protected by Gaussian noise added within the encrypted domain before decryption. We write $v_t = \{s^t(w,D) + G_w\}_{w \in Q_C}$ for the perturbed score vector at iteration $t$.

**Definition G.1.** Let $\lambda$ be the security parameter. A function $\mu(\lambda)$ is negligible, denoted negl($\lambda$), if for every positive polynomial $p(\cdot)$ and all sufficiently large $\lambda$, $\mu(\lambda) < 1/p(\lambda)$. Let $\approx_c$ denote computational indistinguishability.

**Theorem G.2.** *Assume the underlying FHE scheme $\Pi_{FHE} = (KeyGen, Enc, Dec, Eval)$ is IND-CPA secure. Assuming non-collusion between the CE and CSE, the FHAIM protocol securely computes the DP-synthetic data generation in the presence of static, semi-honest adversaries.*

*Proof.* We prove security via the real/ideal paradigm. We define an ideal functionality $\mathcal{F}$ and construct probabilistic polynomial-time (PPT) simulators $\mathcal{S}_{CE}$ and $\mathcal{S}_{CSE}$ such that, for any PPT adversary $\mathcal{A}$, the simulated view is computationally indistinguishable from the real protocol execution view.

**Ideal Functionality $\mathcal{F}$.** $\mathcal{F}$ receives the private dataset $D$ from DE and internally executes the full AIM algorithm with DP noise. It provides the following outputs to each party:

- **To DE**: the final synthetic dataset $\hat{D}$.

- **To CE**: (i) the selected indices $\{w_t^*\}_{t=1}^T$, and (ii) the noisy marginals $\{\tilde{q}_j\}_{j=1}^M$ (covering both initialization and measurement phases).

- **To CSE**: (i) the plaintext perturbed score vectors $\{\mathbf{v}_t\}_{t=1}^T$, and (ii) the noisy marginals $\{\tilde{q}_j\}_{j=1}^M$.

1. **Simulating the Computation Entity ($\mathcal{S}_{CE}$).**

   The real view of the CE is:
   $$\text{View}_{CE}^{\text{real}} = \left(1^\lambda, \llbracket D \rrbracket_{pk}, \{\llbracket \mathbf{v}_t \rrbracket_{pk}\}_{t=1}^T, \{w_t^*\}_{t=1}^T, \{\tilde{q}_j\}_{j=1}^M\right).$$

   We construct $\mathcal{S}_{CE}$ given the output of $\mathcal{F}$ to CE, namely $\left(\{w_t^*\}_t, \{\tilde{q}_j\}_j\right)$:

   - $\mathcal{S}_{CE}$ generates a key pair $(pk, sk) \leftarrow \text{KeyGen}(1^\lambda)$.
   - It constructs a dummy dataset $D' = \{0\}^{|D|}$ and computes $\llbracket D' \rrbracket_{pk} \leftarrow \text{Enc}(pk, D')$.
   - For each iteration $t$, it constructs dummy score vectors $\mathbf{v}_t' \in \mathbb{R}^{|Q_C|}$ (e.g., all zeros) and encrypts them to obtain $\llbracket \mathbf{v}_t' \rrbracket_{pk}$.
   - It outputs the simulated view:
   $$\text{View}_{CE}^{\text{sim}} = \left(1^\lambda, \llbracket D' \rrbracket_{pk}, \{\llbracket \mathbf{v}_t' \rrbracket_{pk}\}_{t=1}^T, \{w_t^*\}_{t=1}^T, \{\tilde{q}_j\}_{j=1}^M\right).$$

   **Claim:** $\text{View}_{CE}^{\text{real}} \approx_c \text{View}_{CE}^{\text{sim}}$.

   *Proof of Claim:* The plaintext components ($\{w_t^*\}_t$ and $\{\tilde{q}_j\}_j$) are identical in both views, as they are provided directly to the simulator by $\mathcal{F}$. The only difference lies in the underlying plaintexts of the ciphertexts i.e., the real view contains encryptions of $D$ and $\{\mathbf{v}_t\}_t$, while the simulated view contains encryptions of $D'$ and $\{\mathbf{v}_t'\}_t$.

   Crucially, under our semi-honest model, the CSE acts as a **Restricted Decryption Oracle** (see below). The CE only submits legitimate, protocol-gated ciphertexts and cannot adaptively mount a chosen-ciphertext attack.

   Thus, the IND-CPA security of $\Pi_{FHE}$ is sufficient. By definition, $\text{Enc}(pk, m_0) \approx_c \text{Enc}(pk, m_1)$. A standard hybrid argument over all ciphertexts (replacing one real ciphertext with a dummy ciphertext at a time) shows that no PPT adversary can distinguish the two views with more than negligible advantage.

2. **Simulating the Crypto-Service Entity ($\mathcal{S}_{CSE}$).**

   The real view of the CSE is:
   $$\text{View}_{CSE}^{\text{real}} = \left(1^\lambda, sk, \{\mathbf{v}_t\}_{t=1}^T, \{\tilde{q}_j\}_{j=1}^M\right).$$

   That is, beyond its own key material $(1^\lambda, sk)$, the CSE observes exactly the plaintext results of its decryption operations: the perturbed score vectors and the noisy marginals.

   We construct $\mathcal{S}_{CSE}$ given $(sk)$ and the output of $\mathcal{F}$ to CSE, namely $\left(\{\mathbf{v}_t\}_t, \{\tilde{q}_j\}_j\right)$:

- $\mathcal{S}_{CSE}$ directly outputs:

$$\text{View}_{CSE}^{\text{sim}} = \left(1^\lambda,\, sk,\, \{\mathbf{v}_t\}_{t=1}^T,\, \{\tilde{q}_j\}_{j=1}^M\right).$$

**Claim:** $\text{View}_{CSE}^{\text{real}} \equiv \text{View}_{CSE}^{\text{sim}}$.

*Proof of Claim:* In the real protocol, the CSE's view beyond $(1^\lambda, sk)$ consists exactly of the decrypted outputs: the perturbed score vectors $\{\mathbf{v}_t\}_t$ (from $\pi_{\text{SELECT}}$) and the noisy marginals $\{\tilde{q}_j\}_j$ (from initialization and $\pi_{\text{MEASURE}}$). Since $\mathcal{F}$ provides these same values to $\mathcal{S}_{CSE}$, the simulated view is identical to the real view. □

**Conclusion.** Since both simulators successfully produce views indistinguishable from (resp. identical to) the real execution using only the designated outputs of $\mathcal{F}$, and since $\mathcal{F}$ does not reveal the raw data $D$ or unperturbed statistics to either party, the protocol is secure under the non-collusion assumption. □

□

**Restricted Decryption Oracle.** The underlying FHE scheme (e.g., CKKS) provides IND-CPA security. However, because the CE sends ciphertexts to the CSE for decryption during the select and measure steps, the CSE effectively acts as a decryption oracle. Generally, unrestricted access to a decryption oracle compromises IND-CPA security. We explicitly address this by defining the CSE as a Restricted Decryption Oracle:

1. **Input Restriction**: The CSE is programmatically restricted to only accept ciphertexts tagged as noise-perturbed outputs: perturbed score vectors $[\![\mathbf{v}_t]\!]$ and noisy marginals $[\![\tilde{q}_j]\!]$.

2. **Output Restriction**: For perturbed score vectors, the CSE does not return the full decrypted vector to the CE; it computes $\arg\max$ and returns only the selected index $w_t^*$. For noisy marginals, the CSE returns the decrypted values to the CE; this is permissible because the marginals are already protected by DP noise added within the encrypted domain.

Because the CE cannot use the CSE to decrypt arbitrary ciphertexts, the CE cannot mount a Chosen Ciphertext Attack. Therefore, the cryptographic security of the data held by the CE remains fully bounded by the IND-CPA guarantees of the FHE scheme.

**Theorem G.3.** *The view of the untrusted Computation Entity (View$_{CE}$) satisfies $(\epsilon, \delta)$-differential privacy.*

*Proof.* The CE's view consists of encrypted states and plaintext outputs received from the CSE. We analyze each component:

1. **Encrypted states**: Under the IND-CPA security of the FHE scheme, all ciphertexts including $[\![D]\!]$, intermediate encrypted computations, and encrypted noise-perturbed values are computationally indistinguishable from encryptions of zero. These contribute zero to the privacy loss budget.

2. **Plaintext outputs**: The CE receives two categories of plaintext information:

   (a) **Selected indices**: Each $w_t^* = \arg\max_{w \in Q_C}(s^t(w,D) + G_w)$, where $G_w \sim \text{Gumbel}(0, \frac{2\Delta}{\epsilon})$, is the output of the Report Noisy Max mechanism applied with global sensitivity $\Delta = \max_{w \in W} \alpha_w(2N+1)$ (Theorem D.1). Each selection satisfies $\frac{\epsilon}{2}$-DP (equivalently, $\frac{\epsilon^2}{8}$-zCDP).

   (b) **Noisy marginals**: Each $\tilde{q}_j = q_{w_j}(D) + \eta_j$ with $\eta_j \sim \mathcal{N}(0, \sigma_j^2 I_{\omega_{w_j}})$ is the output of the Gaussian mechanism with sensitivity 1. This covers both the $d$ initialization marginals (with noise scale $\sigma_{\text{init}}$, each contributing $\frac{1}{2\sigma_{\text{init}}^2}$-zCDP) and the $T$ iterative measured marginals (with noise scale $\sigma_t$, each contributing $\frac{1}{2\sigma_t^2}$-zCDP).

3. **Composition**: The total privacy cost over all plaintext outputs is bounded by zCDP composition (the zCDP parameters of all mechanisms above are summed), and is subsequently converted to an overall $(\epsilon, \delta)$-DP guarantee following the standard zCDP-to-DP conversion (as in AIM).

4. **Post-Processing**: By the Post-Processing property of DP, any subsequent computation the CE performs using the DP-protected outputs, including the probabilistic graphical model updates in the generate step, cannot increase the privacy loss.

$\square$

*Note on the CSE's View.* The CSE resides within the Data Owner's trust boundary. Unlike the CE, the CSE observes the full perturbed score vectors $\mathbf{v}_t$ (not merely the indices $w_t^*$), in addition to the noisy marginals. However, all values observed by the CSE are noise-perturbed: the DP noise is injected by the DE within the encrypted domain prior to decryption. The raw, unperturbed quality scores $s^t(w, D)$ and the true marginals $q_w(D)$ are never exposed in plaintext to any entity, including the CSE.

