# OpenReview forum: "$\texttt{FHAIM}$: Fully Homomorphic AIM for Private Tabular Synthetic Data Generation"
_ICML.cc/2026/Conference — ICML 2026 regular_

### Official Review · Reviewer_mZRk · 2026-02-19

**Soundness:** 3
**Presentation:** 3
**Significance:** 3
**Originality:** 3
**Overall Recommendation:** 5
**Confidence:** 4

**Summary:**

This work designs and implements an FHE-based protocol for AIM (Adaptive and Iterative Mechanism), a type of SDG (Synthetic Data Generation). SDG applies differential privacy (DP) to the original dataset to generate a new synthetic dataset that preserves similar statistical properties, with the goal of protecting privacy. However, SDG is often difficult to deploy in practice, which has motivated SDG-as-a-service offerings. In such a service model, the data provider must reveal the original data to the server, creating an additional privacy risk. The authors argue that FHE can mitigate this issue by enabling SDG without exposing the raw data to the server. They propose a three-party protocol involving an FHE-key provider, a data provider, and a computing entity, where most of the computational burden is placed on the computing entity. Their system generates AIM-based synthetic data under a semi-honest threat model while achieving DP as the privacy goal.

**Compliance With Llm Reviewing Policy:**

Affirmed.

**Final Justification:**

I was previously uncertain about the paper’s 4-party construction. However, I have now confirmed that it is in fact essentially a conventional user-server 2-party construction, and accordingly I raise both my confidence level and my score.

**Key Questions For Authors:**

(1) Which CKKS parameters and library did you use?
(2) Roughly, what is the value of $k$? If $k$ is small, it seems that somewhat homomorphic encryption might also be feasible.
(3) What is the ciphertext size that the DE needs to send? When using OHE, it seems to be quite large.

**Limitations:**

yes

**Strengths And Weaknesses:**

Strength:
The problem statement in the introduction about the privacy issue inherent in SDG-as-a-service (what the authors call input privacy) is compelling and increases the paper’s significance. The proposed direction—that FHE can address this issue—is plausible. The observations that (i) MPC-based solutions require a non-collusion assumption and (ii) prior FL-based work does not solve the single data holder–service provider setting serve as useful reference points that motivate FHE as an alternative. Introducing FHE to this problem is also a positive aspect in terms of originality. The paper is generally well structured and well written, with good readability.

Weakness:
This submission adopts a three-party formulation consisting of the Data Entity (DE), Crypto-Service Entity (CSE), and Computation Entity (CE), where the CE is identified with the Generation Entity. The claimed guarantees rely on the non-collusion assumption between the CSE and the CE, and the CSE performs lightweight but non-cryptographic operations such as $\arg\max$. This weakens the authors' claim that their approach is meaningfully differentiated from MPC-based SDG-aaS.

According to the current protocol, the DE’s intended security goal seems to be that both the CSE and the CE should only ever learn DP-sanitized information. It would be helpful if the authors stated this goal explicitly and precisely, since it can be easily conflated with the indistinguishability-based security notion provided by FHE.
From the viewpoint that the CE effectively receives access to a decryption oracle through the CSE, the setting seems to require IND-CPA$^D$ security. It would be beneficial to briefly acknowledge and discuss this point.

From the perspective of optimizing FHE computation, the SIMD utilization also appears inefficient. A single CKKS ciphertext (which can store up to roughly $2^{14}\sim 2^{16}$ values depending on parameters) contains only $N$(=228, 4120, 614 in the experiments), which likely induces substantial wasted capacity. There should be alternative designs that better exploit CKKS SIMD, e.g., bit-reversal vector encoding methods that allow using most slots. (e.g. four 4-vectors $(a_0,...,a_3),...,(d_0,...,d_3)$ can be encoded to a 16-vector $(a_0,b_0,c_0,d_0,...,a_3,b_3,c_3,d_3)$.) This could potentially enable encrypting more attributes per ciphertext and significantly reduce overall latency.

---

> ### Author Rebuttal · Authors · 2026-03-31
>
> Thank you for your detailed feedback and for considering our work as compelling and original. We address your comments below to clarify the technical details and strengthen the case for the paper.
>
> * **Response to Questions.** We provide the details of these parameters below and will include them in the final version of the paper as well for clarity and completeness.
>     1. We use OpenFHE with CKKS-RNS. The parameters are: multiplicative depth 10, scaling modulus 59 bits, first modulus 60 bits (total modulus 650 bits), and 128-bit security, giving ring dimension $N=2^{15}$ and batch size 16,384. The multiplicative depth is set to 10 rather than the minimum required ($\approx2$ for FHAIM-L2, $\approx5$ for FHAIM-L1) for two reasons: CKKS approximate arithmetic benefits from extra levels as a precision buffer, and OpenFHE's internal EvalPoly implementation may consume additional levels beyond the theoretical minimum. No bootstrapping is used so our implementation operates as leveled FHE throughout.
>     2. Since in our experiments we consider only 1-way and 2-way marginals, we have set $k=2$. Your intuition is correct about feasibility of somewhat HE. In our implementation we use CKKS scheme **without** bootstrapping. This is precisely leveled FHE (SHE): it supports a fixed bounded number of multiplications with no bootstrapping. The reason we use CKKS rather than other SHE schemes (BGV/BFV) is that DP noise — Gaussian samples in the MEASURE step and Gumbel samples in the SELECT step involves real-valued arithmetic. BGV and BFV operate over integers and would require discretizing continuous noise distributions, introducing quantization errors that interact poorly with DP guarantees. CKKS natively supports real-valued approximate arithmetic, making it the natural choice regardless of depth requirements.
>     3. The DE uploads 45, 24, and 39 ciphertexts ($\approx270 MB, \approx144 MB, \approx234 MB$) for Cancer, COMPAS, and Diabetes respectively, plus a small number of noise ciphertexts. Although OHE can appear large, our column-wise SIMD packing encodes all records into a single ciphertext per OHE column, so the upload scales with total domain size $\sum \omega_i$ rather than dataset size $N$  and is a one-time cost after which DE goes fully offline.
>
> * **On packing strategy.** Under column-wise packing, $InnerProduct(ct_i, ct_j)$ naturally performs a single $\texttt{EvalMult}(ct_i, ct_j)$ followed by a contiguous $\texttt{EvalSum}$ ($O(\log N)$ standard rotations).
>
>     Under interleaved packing, each inner product requires additional steps: a rotation to align attributes, a plaintext mask to zero out garbage slots, and a strided $\texttt{EvalSum}$ requiring non-standard rotation keys. These penalties are paid for every one of the $\sum_{i<j}\omega_i\omega_j$ inner products in the **Compute** step. We empirically verified this using OpenFHE across $N \in \{256, 512, 1024\}$, finding column-wise packing to be **1.5-1.7× faster**. Moreover, interleaved packing becomes infeasible for $N \geq 2048$ since the flattened dimension $N \cdot \sum \omega_i$ exceeds the CKKS batch size limit, making column-wise packing both faster and more scalable for FHAIM.
>
>
> * **On security:** The DE's security goal is that the CE learns only $(\epsilon,\delta)$-DP outputs and never observes raw data or unperturbed statistics.
>
>     As we detailed in our response to Reviewer X1NH, the DE and CSE are logical roles. In a standard deployment, Alice may act as both the DE and the CSE (i.e., Alice manages her own keys). In this Client-Server model, the CE does not have access to an independent ''decryption oracle." The CE is simply returning the intermediate, DP-noised outputs back to the client (Alice) for protocol continuation. Furthermore, because the CE operates under a semi-honest threat model, it strictly follows the protocol and will only ever submit legitimate, DP-sanitized ciphertexts to Alice. It will not adaptively construct malicious ciphertexts to exploit her decryption capabilities (which would require IND-CCA security). Because the CE's interactions are strictly bounded by honest protocol execution, IND-CPA security is mathematically sufficient. Please refer to the security proof sketch in response to reviewer 66ng. We will briefly acknowledge and discuss this threat-model distinction in the revised manuscript.
>
>
>
> * **On Trust model:** Our non-collusion assumption is qualitatively different from MPC's. In MPC, non-collusion is a cryptographic precondition, whereas in FHAIM the non-collusion assumption reduces to key non-disclosure. Please refer to our clarification to reviewer X1NH for a more detailed response.

---

> > ### Author Rebuttal · Reviewer_mZRk · 2026-04-02
> >
> > \begin{enumerate}
> >
> > \item CKKS parameter choice, library, leveled-HE setting, and related implementation details are now substantially clarified.
> >
> > \item My concern about underutilization of CKKS slots is only partially resolved, but I view this mostly as an engineering refinement rather than a central issue.
> >
> > \item The ciphertext upload size may still be a practical drawback. That said, this seems more like a performance tradeoff than a conceptual weakness, and there may be alternative FHE design choices that mitigate it, possibly with different tradeoffs.
> >
> > \item I now understand that the 4-party description in the paper is mainly a deployment-level presentation, while the underlying cryptographic structure is essentially 2-party. However, I believe this distinction should be made much more explicit in the paper. The 4-party view is best understood as a real-world deployment of an underlying secure 2-party construction, and the corresponding trust assumptions should be stated clearly and justified carefully.
> >
> > \item My main remaining concern is the security discussion. I firmly maintain that IND-CPA-D remains a relevant notion here. This is weaker than full IND-CCA, but stronger than plain IND-CPA: even when the CE behaves honestly, it still receives decryption results on ciphertexts that it computed and uses them to continue the SDG process. This kind of security concern is well known in FHE, and many HE works explicitly discuss parameters or design choices needed to address IND-CPA-D-type security. Since this paper is not primarily about this security concern itself, it is too much to demand a full treatment. However, I believe the paper should at least explicitly acknowledge and discuss this point. In that sense, the setting involves a passive decryption-feedback channel, and plain IND-CPA alone does not seem to fully capture the relevant security discussion. I refer the authors "On the Security of Homomorphic Encryption on Approximate Numbers" for the concept of ind-cpa-d security.
> >
> > \end{enumerate}
> >
> > On the other hand, One reason for my lower confidence was that I was not fully convinced about the paper’s 4-party model. I have now confirmed that it is essentially a 2-party construction, and I raise both my confidence level and my score.

---

> > > ### Author Response · Authors · 2026-04-08
> > >
> > > We are glad that we could answer your queries and increase your confidence in our work. We also appreciate the increase in the score.
> > >
> > > **On IND-CPA-D**: We thank you for pointing out this relevent work to us. The authors have gone through it and we agree that because the Computation Entity (CE) receives decrypted noisy marginals from the CSE (in the measure step) to continue the iterative generation loop, the system formally operates in an IND-CPA-D setting. We also acknowledge the specific vulnerabilities of the CKKS scheme in this setting, as demonstrated by Li \& Micciancio.
> > >
> > > We will add a dedicated paragraph to the Security/Trust Model section of the paper explicitly acknowledging the IND-CPA-D setting.

---

### Official Review · Reviewer_66ng · 2026-03-02

**Soundness:** 2
**Presentation:** 3
**Significance:** 4
**Originality:** 4
**Overall Recommendation:** 5
**Confidence:** 4

**Summary:**

Aims to use FHE for the process of generating synthetic data. In particular, the paper seeks to address the use case in which a user wants to generate a synthetic dataset from their data for wider distribution but does not want to create the dataset themselves. As well, the synthetic data generation algorithm should be hidden from the data provider. To make this possible, the data provider sends the FHE-encrypted data to the SDG algorithm, which is then combined with the DP.

**Compliance With Llm Reviewing Policy:**

Affirmed.

**Final Justification:**

All concerns were addressed. I will maintain my current score.

**Key Questions For Authors:**

1. Why is there no security proof?

2. Why is there no differential privacy guarantees?

3. Why is the CE separated from the GE? Is there a good reason not to put them together?

**Limitations:**

There is no formal security proof verifying the protocol's security. I understand that security may be guaranteed by the underlying security primitives, but showing this formally is important.

**Strengths And Weaknesses:**

The idea for this paper is highly original and has significant significance in the field. The experiments demonstrate the framework's efficacy well.

The paper moves from the protocol description to the protocol results. Analysis of the DP noise injection and security on a formal level is needed.

---

> ### Author Rebuttal · Authors · 2026-03-31
>
> Thank you for your evaluation and deeming our work to be highly original. We address your comments below to strengthen the contributions of the paper.
>
> * **On security proof (Q1).** We have a complete formal security proof and will add it to the final version. The proof establishes security via real/ideal simulation under the non-collusion assumption:
>     1. For the CE, all data-dependent values remain encrypted; the simulator replaces them with encryptions of zeros, and indistinguishability follows from IND-CPA security via a standard hybrid argument.
>     2. For the CSE (trusted, within DE's boundary), the view consists exactly of the decrypted noise-perturbed values, which the simulator reproduces perfectly given the secret key.
>     3. Because the CE sends ciphertexts to the CSE for decryption, the CSE effectively acts as a decryption oracle. We address this by defining the CSE as a *Restricted Decryption Oracle*: it only accepts ciphertexts tagged as perturbed outputs and, for score vectors, returns only the argmax index. This prevents chosen-ciphertext attacks, so IND-CPA security suffices.
>
> * **On DP(Q2).** The end-to-end DP guarantee follows directly from AIM’s zCDP accounting and Report Noisy Max, unchanged by encrypted execution due to post-processing.
>
> * **On trust model clarification (Q3).** We separate the GE and CE to reflect the real-world decoupling of proprietary algorithms and computational infrastructure. While a single service provider often fulfills both roles (see Line 255), this logical distinction allows a specialized SDG provider (GE such as MDClone) to utilize high-performance cloud resources (CE such as AWS) for FHE execution without compromising their intellectual property. This facilitates a vendor-agnostic deployment model, scaling of compute and establishes separation of powers. So we do not put them together to ensure our framework remains flexible enough to accommodate scenarios where the algorithm designer and the infrastructure provider are different entities.

---

> > ### Author Rebuttal · Reviewer_66ng · 2026-04-03
> >
> > All concerns have been addressed to my satisfaction.

---

> > > ### Author Response · Authors · 2026-04-05
> > >
> > > We appreciate your understanding of our work and your constructive feedback on the security proof. We will include it in the final version of the manuscript

---

### Official Review · Reviewer_X1NH · 2026-03-12

**Soundness:** 3
**Presentation:** 3
**Significance:** 3
**Originality:** 3
**Overall Recommendation:** 4
**Confidence:** 2

**Summary:**

This paper presents FHAIM, an FHE-based extension of AIM for input-private tabular synthetic data generation in an outsourced setting. The method keeps the data-dependent AIM steps in the encrypted domain, introduces FHE protocols for compute/select/measure, and performs generation in the clear only after decrypting already-DP-protected statistics. Empirically, the paper evaluates on three tabular datasets and reports feasible runtimes with utility close to plaintext AIM-L2.

**Compliance With Llm Reviewing Policy:**

Affirmed.

**Final Justification:**

I think the paper needs a clearer statement on the trust model, and the practical applications that may adopt the proposed framework. Overall, I'm fine with this paper and happy to keep my score.

**Key Questions For Authors:**

Q1. The paper emphasizes that it avoids the need for multiple non-colluding parties, but the system still seems to rely on CE and CSE being separate and non-colluding. Could you explain this trust model more plainly, and make the difference from MPC-style assumptions fully explicit?

Q2. Could you clarify more concretely what the CSE is allowed to see and do? Since it has the decryption key, it would help to say clearly what stops it from decrypting more than the intended DP-protected outputs, beyond simply assuming semi-honest behavior.

Q3. I found the description around Protocol 2 a bit hard to follow. Who exactly decrypts the noisy scores, and who performs the final argmax selection? Please make the execution flow and trust boundary completely explicit.

Q4. The paper says the system can work for any SDG algorithm, but the actual construction here looks quite tied to AIM and tabular marginals. Could you separate more clearly what is generic in the framework and what is AIM-specific?

Q5. Could you give a bit more details of scalability beyond the current experiments, for example for larger domain sizes, richer workloads, or downstream evaluations beyond logistic regression? Does the computational cost grow exponentially as the task complexity rise?

**Limitations:**

Yes.

**Strengths And Weaknesses:**

Strengths:

+The paper targets outsourced synthetic **tabular** data generation for a single provider and a single data holder, which is practically meaningful and clearly motivated.

+The contribution goes beyond simply encrypting AIM inputs: it redesigns key data-dependent stages as FHE protocols and addresses practical issues such as encrypted marginal computation and noise handling.

+The experiments suggest the approach is not merely conceptual and can run at moderate scale with acceptable utility loss relative to plaintext baselines.

Weaknesses:

- The main novelty is not really in the ML method itself. The synthesizer is still AIM, so the contribution is much more about cryptographic and systems adaptation than about a new SDG method. I do not think that is fatal, but the paper should position itself more carefully on this point.

- The trust assumptions need to be stated more directly. The authors clearly say "FHAIM, ..., providing input privacy without requiring multiple non-colluding parties" but FHAIM actually do rely on a third party CSE. The framing sometimes makes the setup sound simpler than it really is, since the system still depends on a separation between CE and CSE and on a non-collusion assumption. That distinction matters and should be made very explicit.

- Similarly, I believe the title should contain 'tabular' because FHAIM cannot be generalized to other data types directly.

---

> ### Author Rebuttal · Authors · 2026-03-31
>
> Thank you for your thoughtful feedback and for recognizing our work as practically meaningful and clearly motivated. We respond to your comments below to further clarify the contribution and impact of the paper.
>
> * We agree that FHAIM's primary novelty lies in cryptographic and systems adaptation rather than proposing a new SDG algorithm. We will revise the introduction to emphasize this positioning.
>
> * **On non-collusion and trust model (Q1--Q3).** We clarify the trust model explicitly below and will include it in the revised version.
>     1. CE is untrusted and only learns DP outputs.
>     2. CSE holds the secret key and is within DE's trust boundary. It receives only ciphertexts from CE (noisy marginals during INIT/MEASURE, perturbed scores during SELECT) and returns either decrypted noisy marginals or the argmax index, maintaining the CE's $(\epsilon,\delta)$-DP view.
>     3. No cryptographic non-collusion is required beyond key non-disclosure.
>
> * **On non-collusion.** While FHAIM defines distinct logical roles (DE, CSE, CE, GE) for modularity, in practice, this collapses to a standard 2-Party Client-Server architecture. If the DE (Alice) retains her own secret key and acts as her own decryption oracle, the CSE role is absorbed by the client. There is no third party. Because of this, our 'non-collusion' assumption is fundamentally different from MPC. In MPC, non-collusion is a structural cryptographic requirement where multiple compute servers must be trusted not to combine their data shares. In FHAIM, privacy is rooted entirely in the cryptographic location of the secret key. Our non-collusion assumption simply means the Client (Alice) does not hand her secret key to the Untrusted Server (Bob). Even if the Server controls arbitrary compute infrastructure, it cannot reconstruct the plaintext without the client's key.
>
> * **On CSE and Protocol 2.** To address the concern regarding what stops CSE from 'decrypting more than intended', we consider two scenarios: (1) Alice trusts CSE: Here CSE's information flow is restricted by CE.
> For example, in Protocol 2, CE sends over encrypted value (encrypted perturbed scores for RNM) to CSE , CSE decrypts them (plaintext perturbed scores), performs the required steps (argmax in-the-clear) and sends back only the DP output to CE (DP selected index) (please see response to reviewer 6qmQ). Similarly, in the initialization and measurement steps, CSE receives and decrypts only DP-noised marginals. Thus, maintaining $(\epsilon,\delta)$-DP view for CE. While we do assume a semi-honest behavior for CSE, its ability to learn is limited by CE's role and its incapability to see any plaintext based on information flow. (2) Alice as CSE: When Alice(DE) chooses to perform the functions of CSE herself, the trust model reduces to the standard Client-Server model. In this case, there is no "leakage" to CSE as Alice already owns the raw data.
>
> * **On generalization of the proposed approach (Q4).**
>     - The proposed DP-in-FHE workflow are generic to any SDG that computes data-dependent statistics.  $\pi_{\mathsf{COMP}}$ and $\pi_{\mathsf{MEASURE}}$ are generic for tabular marginals-based algorithms; $\pi_{\mathsf{SELECT}}$ is AIM-specific. For example, extending to MST would require redesigning $\pi_{\mathsf{SELECT}}$ to compute mutual information scores, which involves developing efficient FHE protocols for exp and logsumexp operations.
>     - While AIM is used as the backbone for synthetic text generation as well [1] we agree with the reviewer to revise the title to explicitly mention tabular data.
>
> * **On scalability (Q5).** he computational cost of FHAIM does \emph{not} grow exponentially with task complexity. It scales polynomially in all practically relevant dimensions: (1)**Records $N$**: strictly $O(N/L)$ linear scaling, as established in Section 6 and confirmed by our Adult experiment (see reviewer 6qmQ); (2)**Attributes $d$**: the COMPUTE step scales as $O(d^2 \cdot \bar{\omega}^2 \cdot N/L)$ and the SELECT step as $O(d^3)$; both polynomial; (3)**Domain sizes $\omega_i$**: the 2-way compute cost scales as $\sum_{i<j}\omega_i\omega_j \approx O((\sum_i \omega_i)^2)$, i.e., quadratic in total OHE columns, manageable via domain compression as demonstrated on the Adult dataset; (4)**Downstream tasks:** FHAIM's FHE computation ends at synthetic data generation $\hat{D}$; all downstream evaluations (logistic regression, SVM, neural networks, etc.) operate on plaintext $\hat{D}$ and are entirely decoupled from FHAIM's computational cost.
>
> [1] Y. Hu, R. McKenna, D. Yu, S. Wu, H. Zhao, Z. Xu, P. Kairouz (2025). ACTG-ARL: Differentially Private Conditional Text Generation with RL-Boosted Control. arXiv preprint arXiv:2510.18232.

---

> > ### Author Rebuttal · Reviewer_X1NH · 2026-04-01
> >
> > My main concerns have been addressed. I will keep the current score.

---

> > > ### Author Response · Authors · 2026-04-05
> > >
> > > We thank the reviewer again for their time and consideration and are glad that we could address all their concerns.
> > >
> > > We will carefully revise our manuscript as per your suggestions to avoid any confusion and make the following changes:
> > >    - Clearly state the trust model and be explicit about what each entity is able to see and do.
> > >    - Improve readability of Protocol 2 by mentioning which entity is performing which step
> > >    - Clarify the generic and specific parts of FHAIM for SDG.

---

### Official Review · Reviewer_6qmQ · 2026-03-12

**Soundness:** 2
**Presentation:** 2
**Significance:** 2
**Originality:** 2
**Overall Recommendation:** 3
**Confidence:** 2

**Summary:**

The paper proposes **FHAIM**, a framework that adapts the AIM algorithm for differentially private (DP) tabular data generation to the fully homomorphic encryption (FHE) setting.

In this architecture a single data holder uploads an encrypted dataset to an untrusted server; the server executes the synthetic‑data generation process on ciphertexts and never sees any plaintext. All leakage is therefore constrained by formal DP guarantees, protecting both input and output privacy.

**Compliance With Llm Reviewing Policy:**

Affirmed.

**Final Justification:**

I appreciate the authors' detailed rebuttal, particularly the clarification regarding the system's trust boundaries. However, the explanation provided in the rebuttal (that the CSE is fully within the data holder's trust boundary) seems to contradict the explicit claims of the submitted manuscript.

Specifically, the paper explicitly models the CSE as an adversary:
"Trust Model. We consider CE and CSE to be semi-honest, i.e., they follow the protocols but attempt to learn about D" (Section 4).

Additionally, the paper states:

"Any data seen by the CSE is already protected by DP noise added within the encrypted domain."

"Because the DP noise is applied within the encrypted domain, the CSE can safely decrypt the scores..."

The current manuscript suggests that the CSE is not that trusted.

Hence, I maintain my score.

**Key Questions For Authors:**

Did you test your algorithm on the original datasets used by the AIM paper?

**Limitations:**

See **Strengths And Weaknesses**

**Strengths And Weaknesses:**

The main novelty is the stronger security model. Prior work on private synthetic‑data generation relied on secure multi‑party computation (MPC) with multiple non‑colluding servers --- an assumption that is hard to verify in practice and limits deployability.
Switching to FHE removes that requirement. The authors also introduce several optimization tricks to make AIM more practical under FHE.

However, there is a **flaw** in the current protocol description.

- **Protocol 2** (Section 4.2) exposes the *noisy scores* (values perturbed by Gumbel noise) to the untrusted server. Revealing these values violates DP: the server must only learn which index has the maximum noisy score (the arg‑max index), not the scores themselves. To be DP‑compliant the server should operate on encrypted noisy scores and output only an encrypted index, which the client decrypts; only the index is ever revealed in the clear.

On the other hand, evaluating an arg‑max under FHE is expensive.

Some minor comments, which the authors should address in a revision:

1. The interface of the `Combine` procedure in Protocol 1 (main text) does not match the definition in the appendix – the number of arguments differ.
2. Dataset sizes are reported only in the appendix; the main paper should state them explicitly.
   The sizes used here appear much smaller than those in the original AIM experiments.

---

> ### Author Rebuttal · Authors · 2026-03-31
>
> Thank you for your careful evaluation of our paper. We have taken your comments seriously and respond below with additional clarification and justification to address the concerns and  strengthen the case for the paper.
>
> * **On Protocol 2.** The reviewer is correct that revealing the noisy scores to an *untrusted* server would violate DP, but this is not what we do. We do not reveal the noisy scores to the untrusted CE. Instead, the noise scores are decrypted by the CSE, which is either the data holder or its cryptographic delegate. Our DP guarantee is defined with respect to the untrusted compute entity (CE), not with respect to the data holder or trusted CSE, which, as common in the FHE literature, lies inside the privacy boundary. The CSE returns only the index of the selected $w^*$ to the untrusted CE for further computation. Under the post-processing property of DP, disclosing only the index ensures the CE's view remains $(\epsilon,\delta)$-DP. We will update Protocol 2's description to be more explicit in describing this boundary and add the following text:
> > In $\pi_{\mathsf{SELECT}}$, the CE sends the encrypted noisy scores $[\![s_w]\!]$ on Line 10 to the CSE. The CSE, being a trusted entity within the privacy boundary, decrypts the vector (on Line 11) to find the argmax (on Line 13). The CSE sends $w^*$ to CE, ensuring that the CE only receives the DP selected index, maintaining the standard $(\epsilon,\delta)$-DP guarantee of the Report Noisy Max mechanism. FHE protocols for argmax, such as those based on polynomial approximation  (e.g. [1]) or scheme-switching, could in principle be implemented and integrated into FHAIM without altering the underlying workflow. However both these methods would incur both computational and memory cost because of either increased multiplicative depth requirement (in chained polynomial evaluation) or latency of scheme-switching itself. For example, in our experiments on the COMPAS dataset with scheme-switching implementation of argmax, the latency increased by 6x-7x. Adopting such FHE protocols, the role of the CSE can instead be minimized to a simple decryption step for determining the selected index, as illustrated in Figure 2, albeit at a substantial performance cost. In our implementation, we prioritize correctness-under-trust over costly cryptographic operations.
>
> * **On experiments with original datasets in AIM paper (Q1).** Yes, we directly evaluated on the Adult dataset used in the original AIM paper. AIM completed synthetic data generation in 19 mins and 5 mins respectively for AIM-L1 and AIM-L2, while FHAIM required $\sim$ 5.5 hours and $\sim$ 5.3 hours for FHAIM-L1 and FHAIM-L2 respectively (here we used ring dimension = $2^{17}$ to accommodate all records). We will add this information to the paper.
>
> We will address the minor comments related to the appendix in the camera-ready version of the paper.
>
> [1] P. Zhang, A. Duan, and H. Lu. "An efficient homomorphic argmax approximation for privacy-preserving neural networks." Cryptography 8.2 (2024): 18.

---

> > ### Author Rebuttal · Reviewer_6qmQ · 2026-04-03
> >
> > In Section 4, the paper states:
> >
> > > “Any data seen by the CSE is already protected by DP noise added within the encrypted domain.”
> >
> > However, as described, the CSE’s view does not appear to satisfy differential privacy, as revealing Gumbel-noisy values does not provide the intended protection.
> >
> > Could you clarify what security or privacy guarantees apply to the CSE’s view?
> >
> > If the CSE is assumed to be fully trusted, would it be possible to use it to run the synthetic data generation protocol directly?

---

> > > ### Author Response · Authors · 2026-04-05
> > >
> > > - The CSE receives Gumbel-perturbed quality scores  and Gaussian-noised marginals. These perturbed quality scores are by themselves not differentially private. That is not a problem because the CSE is within the trust boundary of the raw data holder (Alice/DE) (as color-coded in Figure 1), and one could even choose DE=CSE.
> > >     The formal security guarantee for the CSE's view is simulation-based (please see our response to Reviewer 66ng), where the CSE is defined as a Restricted Decryption Oracle.
> > >     We will update the statement in Section 4 to precisely state CSE’s security and privacy views.
> > >
> > > - The CSE cannot run the synthetic data generation (SDG) protocol directly because it doesn't know this protocol. If Alice (DE \& CSE) knew how to do SDG then she wouldn't need to outsource it to Bob (GE \& CE). The existence of numerous SDG-as-a-service companies illustrates that there is a market/need for such outsourced SDG, where Alice doesn't have the know-how or the ability to do SDG on her end. The CSE's only role is lightweight decryption within Alice's trust boundary. Everything that leaves Alice's trust boundary (DE+CSE) is either encrypted under FHE or protected by formal DP guarantees.

---

### Decision · Program_Chairs · 2026-04-30

**Decision:**

Accept (regular)

**Comment:**

The majority of reviewers agreed that the paper is significant because it presents a novel three-party protocol for generating AIM-based synthetic data under a semi-honest threat model while achieving DP. There are multiple strengths described by these reviewers: (a) The contribution goes beyond simply encrypting AIM inputs: it redesigns key data-dependent stages as FHE protocols and addresses practical issues such as encrypted marginal computation and noise handling; (b) The experiments suggest the approach is not merely conceptual and can run at moderate scale with acceptable utility loss relative to plaintext baselines; (c) Introducing FHE to this problem is also a positive aspect in terms of originality; and (d) The paper is generally well structured and well written, with good readability.

However, a reviewer noted that the explanation provided in the rebuttal (that the CSE is fully within the data holder's trust boundary) seems to contradict the explicit claims of the submitted manuscript. This poses a concern, stemming either from a lack of clarity in the rebuttal and submitted paper or solely from the reviewer's confusion about both. In either case, I do not feel strongly about this paper, given the remaining concern.